# ERROR FEEDBACK SHINES WHEN FEATURES ARE RARE

## ABSTRACT

We provide the first proof that gradient descent (GD) with greedy sparsification (TopK) and error feedback (EF) can obtain better communication complexity than vanilla GD when solving the following distributed optimization problem $\min_{x \in \mathbb{R}^d} \{ f(x) = \frac{1}{n} \sum_{i=1}^{n} f_i(x) \}$, where $n$ = # of clients, $d$ = # of features, and $f_1, \ldots, f_n$ are smooth nonconvex functions. Despite intensive research since 2014 when EF was first proposed by Seide et al., this problem remained open until now. Perhaps surprisingly, we show that EF shines in the regime when features are rare, i.e., when each feature is present in the data owned by a small number of clients only. To illustrate our main result, we show that in order to find a random vector $\hat{x}$ such that $\|\nabla f(\hat{x})\|^2 \leq \varepsilon$ in expectation, GD with the Top1 sparsifier and EF requires $\mathcal{O}\left( \left( L + r \sqrt{\frac{c}{n} \min \left\{ \frac{c}{n} \max_i L_i^2, \frac{1}{n} \sum_{i=1}^{n} L_i^2 \right\}} \right) \frac{1}{\varepsilon} \right)$ bits to be communicated by each worker to the server only, where $L$ is the smoothness constant of $f$, $L_i$ is the smoothness constant of $f_i$, $c$ is the maximal number of clients owning any feature ($1 \leq c \leq n$), and $r$ is the maximal number of features owned by any client ($1 \leq r \leq d$). Clearly, the communication complexity improves as $c$ decreases (i.e., as features become more rare), and can be much better than the $\mathcal{O}(rL\frac{1}{\varepsilon})$ communication complexity of GD in the same regime.

## 1 INTRODUCTION

In recent decades, the field of machine learning has undergone significant growth and development, presenting numerous opportunities for innovation and advancement. As a result, breakthroughs have been made in various areas such as computer vision (Krizhevsky et al., 2012; He et al., 2015; Ho et al., 2020), natural language processing (Vaswani et al., 2017; Brown et al., 2020), reinforcement learning (Mnih et al., 2013; Sutton and Barto, 2018), healthcare (Esteva et al., 2019), and finance (Dixon et al., 2020). The surge in machine learning applications has also stimulated the development of associated optimization research.

What unique characteristics has modern machine learning introduced to optimization research? i) Firstly, the design of modern models is inherently complex. For example, contemporary convolutional neural networks are constructed from multiple blocks consisting of diverse types of layers that are arranged hierarchically (LeCun et al., 2015). Consequently, these models are notably *nonconvex* in nature (Choromanska et al., 2015). ii) Secondly, the quantity of data utilized during model training is so extensive that the use of a single computing device is no longer practical. Therefore, it is necessary to distribute the data across multiple computing resources, which raises the question of how to effectively coordinate *distributed* (Yang et al., 2019) model training. Another motivation for distributed training is derived from the federated learning framework (Li et al., 2019), where data is owned by users who are not willing to share it with each other in a cross-device federated learning scenario. In this case, the centralized algorithm must manage the training of several client devices. iii) Thirdly, modern machine learning models have a tendency to expand in size, often containing millions of parameters. For instance, in a distributed setting, during gradient descent, each device must transmit a dense gradient vector consisting of millions of parameters each round of communication (Li et al., 2020). This creates an unmanageable burden on the communication network. Hence, there is a need for techniques that can *reduce the number of bits transmitted* through communication channels while maintaining the convergence of the algorithm.

The first two points, namely the lack of convexity and the distributed setup, provide the rationale for focusing our investigation on the finite-sum minimization problem, presented in the following equation:

$$\min_{x \in \mathbb{R}^d} \left\{ f(x) = \tfrac{1}{n} \sum_{i=1}^n f_i(x) \right\}, \tag{1}$$

where $f_i : \mathbb{R}^d \to \mathbb{R}$ is a smooth nonconvex function representing the local loss function on client $i$, and $n \in \mathbb{N}$ is the number of participating clients in the training process, with $d \in \mathbb{N}$ representing the dimension of the model. The third point, which we address by reducing the communication load via lossy compression of updates, is elaborated in the following section.

## 2 COMMUNICATION COMPRESSION AND ERROR FEEDBACK

### 2.1 REDUCING COMMUNICATION IN DISTRIBUTED LEARNING

A common approach to solving the optimization problem (1) is to utilize Distributed Gradient Descent (DGD), which takes the form:

$$x^{t+1} = x^t - \tfrac{\gamma^t}{n} \sum_{i=1}^n \nabla f_i(x^t), \tag{2}$$

where $\gamma^t > 0$ denotes the learning rate. The orchestration of this method occurs between a master node (a central node that holds no data and is responsible for aggregation actions) and $n$ clients. Prior to the start of the algorithm, an initial iterate $x^0 \in \mathbb{R}^d$ and a learning rate $\gamma^0 > 0$ are selected. At iteration $t$, the master node broadcasts the current iterate $x^t \in \mathbb{R}^d$ to the clients, each client then computes its gradient $\nabla f_i(x^t)$ and sends it back to the master. Finally, the master node aggregates all the gradients and uses them to perform the gradient descent step (2), updating the iterate to $x^{t+1}$. This process is repeated.

While DGD is known to be an optimal algorithm for finding a stationary point with a minimum number of iterations on smooth nonconvex problems (Nesterov et al., 2018), it poses a significant challenge to the communication network. At every communication round, DGD transmits dense gradients of $d$ dimensions to the master. As previously mentioned, this communication load is unacceptable in many practical scenarios. One way to resolve this problem is to apply a contractive compressor to the communicated gradients.

**Definition 1.** A (possibly randomized) mapping $\mathcal{C} : \mathbb{R}^d \to \mathbb{R}^d$ is called a contractive compressor if there exists a constant $\alpha \in (0, 1]$, known as the *contraction parameter*, such that

$$\mathbb{E}\left[\|\mathcal{C}(x) - x\|^2\right] \le (1 - \alpha)\|x\|^2, \quad \forall x \in \mathbb{R}^d. \tag{3}$$

An immensely popular contractive compressor is the TopK operator. Given a vector $x \in \mathbb{R}^d$, this operator retains only the $k$ elements with the largest magnitudes and sets the remaining $d - k$ elements to zeroes. It is well-known that the TopK operator is contractive with $\alpha = \frac{k}{d}$.

The next step in reducing the communication load within DGD, is to implement workers-to-server communication compression. This modification of DGD is known as Distributed Compressed Gradient Descent (DCGD), performing iterations

$$x^{t+1} = x^t - \tfrac{\gamma^t}{n} \sum_{i=1}^n \mathcal{C}_i^t(\nabla f_i(x^t)), \tag{4}$$

where $\mathcal{C}_i^t$ is the compressor used by client $i$ at iteration $t$.

DCGD has been shown to converge to stationary points using certain compressors (Gorbunov et al., 2020; Khirirat et al., 2018). However, greedy contractive compressors such as TopK, which are often preferred in practical applications (Szlendak et al., 2022), are not among them, and their practical superiority is not explained by any existing theoretical results. Furthermore, negative results suggest that DCGD with greedy contractive compressors such as TopK can experience exponential divergence in the distributed setting (Beznosikov et al., 2020; Karimireddy et al., 2019), even on simple convex quadratics. Hence, in order to safely apply contractive compressors in a multi-device communication network, a fix is needed. Fortunately, a fix exists: it is generically known as Error Feedback.

## 2.2 EVOLUTION OF ERROR FEEDBACK

The concept of Error Feedback (EF), also known as Error Compensation, was initially proposed by Seide et al. (2014) as a heuristic approach to fix the divergence issues of DCGD. The first version of EF, which for the purposes of this paper we shall denote as EF14, is structured as follows:

$$v_i^t = \mathcal{C}_i^t(e_i^t + \gamma^t \nabla f_i(x^t)), \quad x^{t+1} = x^t - \frac{1}{n} \sum_{i=1}^n v_i^t, \quad e_i^{t+1} = e_i^t + \gamma^t \nabla f_i(x^t) - v_i^t.$$

Here, $\mathcal{C}_i^t$ represents a contractive compressor utilized by client $i$ at iteration $t$, and $e_i^t$ is a memory vector that preserves all elements that were not transmitted by client $i$ in the preceding iterations. In this approach, at iteration $t$, the method sends compressed gradient compensated by the memory vector. The memory vector is then updated and the process is repeated.

The first attempts to establish a theoretical basis for EF14 focused on the single-node scenario (Stich et al., 2018; Alistarh et al., 2018), which is, however, detached from the practical utility of the mechanism that is meaningful only in the distributed setting. An analysis in the distributed setting was executed by (Beznosikov et al., 2020) for strongly convex problems. However, a key limitation of their result was that the (expected) linear rate was only achievable in the homogeneous over-parameterized setting. Subsequent efforts aimed to relax these assumptions and allow for a fast rate even under data heterogeneity, but the EF14 framework seem too elusive to yield vastly improved results. For example, Koloskova et al. (2020) established convergence of EF14 under the assumption of bounded gradients. The authors demonstrate that the method requires $T = \mathcal{O}(\varepsilon^{-3/2})$ iterations to attain an $\varepsilon$-accuracy in terms of the squared gradient norm. However, this rate is worse than the $T = \mathcal{O}(\varepsilon^{-1})$ rate of vanilla DGD, and under stronger assumptions.

---

**Algorithm 1** EF21: Error Feedback 2021 with the TopK$_i$ compressor (Richtárik et al., 2021)

---

1: **Input:** $x^0 \in \mathbb{R}^d$; $g_1^0 \in \mathbb{R}_1^d, \ldots, g_n^0 \in \mathbb{R}_n^d$ (as defined in equation (10)); stepsize $\gamma > 0$; sparsification levels $K_1, \ldots, K_n \in [d]$; number of iterations $T > 0$
2: **Initialize:** $g^0 = \frac{1}{n} \sum_{i=1}^n g_i^0$
3: **for** $t = 0, 1, 2, \ldots, T - 1$ **do**
4: $\quad x^{t+1} = x^t - \gamma g^t$
5: $\quad$ Broadcast $x^{t+1}$ to the clients
6: $\quad$ **for** $i = 1, \ldots, n$ **on the clients in parallel do**
7: $\quad\quad g_i^{t+1} = g_i^t + \text{TopK}_i(\nabla f_i(x^{t+1}) - g_i^t)$
8: $\quad\quad$ Send $g_i^{t+1}$ to the server
9: $\quad$ **end for**
10: $\quad g^{t+1} = \frac{1}{n} \sum_{i=1}^n g_i^{t+1}$
11: **end for**
12: **Output:** $x^T$

---

Recently, Richtárik et al. (2021) re-engineered the EF technique and proposed their EF21 method. This algorithm, presented in a specific version with the TopK$_i$ compressors in Algorithm 1, achieves the optimal number of iterations of $O(\varepsilon^{-1})$ for smooth nonconvex problems, without invoking any strong assumptions. This paper represents a breakthrough in realizing the complete potential of EF for distributed learning applications since EF21 achieves a better convergence rate than EF14, under weaker assumptions, and exhibits better performance in practice. While it may seem that EF21 constitutes the ultimate culmination of the Error Feedback story, we argue that this is far from true! And this is the starting point of our work. We elaborate on this in the next section.

## 3 ERROR FEEDBACK: DISCREPANCY BETWEEN THEORY AND PRACTICE

Our work is motivated through *three observations* about the unreasonable effectiveness of EF21.

### 3.1 OBSERVATION 1: IN PRACTICE, EF21 CAN BE EXTREMELY GOOD!

Let us start by looking at the practical performance of EF21 using the TopK compressor, contrasted with the performance of the current best method for solving (1) in the smooth nonconvex regime in

terms of theoretical communication complexity: the MARINA method of Gorbunov et al. (2021). In Figure 1, which we adopted from from (Szlendak et al., 2022), EF21 is compared to two variants of MARINA: with the RandK and PermK compressors used by the clients. While TopK is a *greedy* mechanism, performed by each client without any regard for what the other clients do, PermK is the exact opposite: it is a *collaborative* compression mechanism.

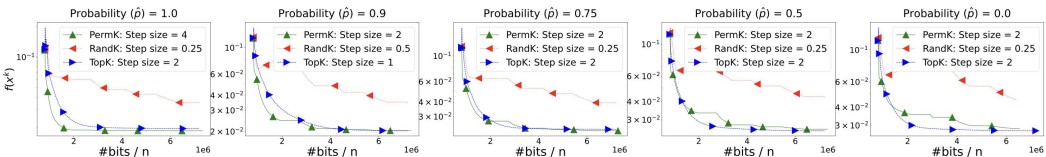

Figure 1: Comparison of three algorithms on the encoding learning task for the MNIST dataset (Szlendak et al., 2022, Figure 2).

It is clear from Figure 1 that EF21 + TopK has similar performance to MARINA + PermK, and both are much faster than MARINA with the "naive" RandK compressor.

### 3.2 OBSERVATION 2: IN THEORY, EF21 IS NOT BETTER THAN DGD!

However—and we believe that this is very important—while the theoretical communication complexity of MARINA is significantly better than that of DGD (Gorbunov et al., 2021), the theoretical communication complexity of EF21 at best matches the complexity of DGD! To state this formally, let us revisit the assumptions Richtárik et al. (2021) used to perform their analysis of EF21.

**Assumption 1.** *The function $f$ is $L$-smooth, i.e., there exists $L > 0$ such that*

$$\|\nabla f(x) - \nabla f(y)\| \leq L \|x - y\|, \quad \forall x, y \in \mathbb{R}^d. \tag{5}$$

**Assumption 2.** *The functions $f_i$ are $L_i$-smooth, i.e., there exists $L_i > 0$ such that*

$$\|\nabla f_i(x) - \nabla f_i(y)\| \leq L_i \|x - y\|, \quad \forall x, y \in \mathbb{R}^d. \tag{6}$$

**Assumption 3.** *There exists $f^* \in \mathbb{R}$ such that that $f(x) \geq f^*$ for all $x \in \mathbb{R}^d$.*

The following observation formalized the above claim.

**Lemma 1.** *Under Assumptions 1, 2, and 3, the optimal theoretical communication complexity of Algorithm 1 obtained by Richtárik et al. (2021) is achieved when the parameter $K$ in the TopK compressor equals the dimension $d$.*

So, to summarize, in theory, EF21 at its best reduces to vanilla DGD, but in practice, EF21 can be orders of magnitude better than DGD. Why is error feedback so unreasonably effective? Despite about a decade of research on the Error Feedback mechanism since the pioneering work of Seide et al. (2014), and steady advances in the field and our theoretical understanding, there is clearly still much to be explored regarding the communication complexity advantages it offers.

### 3.3 OBSERVATION 3: EF21 HAS ABSOLUTELY NO SCALING IN THE HOMOGENEOUS DATA REGIME!

Our final observation is that EF21 performs remarkably poorly in the fully data homogeneous regime, i.e., when $f_1 = f_2 = \cdots = f_n = f$. For simplicity, assume that all initial gradient estimators $g_1^0, \ldots, g_n^0$ are identical. In this case, running EF21 with the TopK compressor in the distributed ($n > 1$) setting is equivalent to running it in the single node ($n = 1$) setting! That is, EF21 has no parallel scaling at all! Specifically, if at iteration $t$ we have $g_i^t \equiv g'$ for all $i$, then $g_i^{t+1} = g_i^t + \text{TopK}(\nabla f_i(x^t) - g_i^t) = g' + \text{TopK}(\nabla f(x^t) - g')$ for all $i$, meaning that all $g_i^{t+1}$ coincide, again. Note that the average of these estimators does not depend on $n$ and, as a result, EF21 generates the same chain of iterates $x^t$ for any value of $n$.

Table 1: Communication complexity of DGD with the Top1 compressor and Error Feedback (either EF14 due to Seide et al. (2014), or EF21 due to Richtárik et al. (2021)). $L$ = smoothness constant of $f$; $L_i$ = smoothness constant of $f_i$; $L_{\max} = \max_i L_i$; $\tilde{L}^2 = \frac{1}{n} \sum_{i=1}^{n} L_i^2$; $L_+$ is the smoothness constant defined in Assumption 4. Note that $L_+ \leq \tilde{L}$ and $L \leq \tilde{L}$.

| Base method | Compressor | Error feedback mechanism | Communication complexity | Better than DGD? |
|---|---|---|---|---|
| DGD | — | — | $\mathcal{O}(rL\varepsilon^{-1})$ | = |
| DGD | Top1 | — | diverges[i] | ✗ |
| DGD | Top1 | EF14 (Seide et al., 2014) | $\mathcal{O}\left(\varepsilon^{-3/2}\right)$ [iii] | ✗ |
| DGD | Top1 | EF21 (Richtárik et al., 2021) | $\mathcal{O}\left(\left(L + r\tilde{L}\right)\varepsilon^{-1}\right)$ [iv] | ✗ |
| DGD | Top1 | EF21 (Richtárik et al., 2021) | $\mathcal{O}\left(\left(L + r\sqrt{\frac{c}{n}}L_+\right)\varepsilon^{-1}\right)$ [v] | ✓[ii] |
| DGD | Top1 | EF21 (Richtárik et al., 2021) | $\mathcal{O}\left(\left(L + r\sqrt{\frac{c}{n}\min\left\{\frac{c}{n}L_{\max}^2, \tilde{L}^2\right\}}\right)\varepsilon^{-1}\right)$ [v] | ✓[ii] |

[i] See Beznosikov et al. (2020); [ii] If $\sqrt{\frac{c}{n}}L_+ \leq L$; [iii] Proved by Koloskova et al. (2020) under bounded gradient assumption; [iv] Proved by Richtárik et al. (2021); [v] **New results proved in this paper.**

### 3.4 SUMMARY OF CONTRIBUTIONS

The above three observations point to deep issues in our theoretical understanding of EF in general, and its most recent and best performing incarnation EF21 in particular. However, our last observation also offers a hint at a possible resolution. Indeed, since EF21 cannot possibly perform well on data homogeneous problems, we conjecture that its practical superiority is due to a certain "favorable" type of data heterogeneity. This is perhaps counter-intuitive because heterogeneous problems are typically much harder than homogeneous ones (Li et al., 2019). Specifically, our contributions are:

**a) Sparsity rules & the first time error feedback beats gradient descent.** We present a refined theoretical analysis of the EF21 algorithm (with the greedy TopK compressor) demonstrating that, under certain conditions on the *sparsity parameters $c$ and $r$* associated with the problem (defined in (9)), EF21 can surpass DGD in terms of *theoretical communication complexity*. This is the first result in the vast literature on error feedback of this type. In particular, in Theorem 2, we provide a stronger convergence rate for EF21 in terms of its communication complexity. For EF21 with the Top1 compressor, for example, we establish an $\mathcal{O}((L + r\sqrt{\frac{c}{n}}L_+)\frac{1}{\varepsilon})$ communication complexity, where $L_+$ is defined in Assumption 4. In the regime when $\sqrt{\frac{c}{n}}L_+ \leq L$, which holds when either $c$ is small or $n$ is large, EF21 is better than DGD.

**b) Experimental validation.** We conduct rigorous toy experiments on linear regression functions to validate our theoretical findings. In one of our experiments, we redefine $c$ as an adaptive parameter $c^t$ satisfying the inequality $\|g^t - \nabla f(x^t)\|^2 \leq \frac{c^t}{n}G^t$ presented in Lemma 5. We also introduce a heuristically defined adaptive stepsize $\gamma(c^t)$, which demonstrates good performance in a logistic regression experiment on non-sparse data.

We believe that this paper represents a milestone in advancing our understanding of Error Feedback algorithms, and lays the groundwork for further advancements in the field.

## 4 RARE FEATURES

We begin our narrative by delineating the key features of the sparsity pattern in $f_i$. Let us define

$$\mathcal{Z} \overset{\text{def}}{=} \left\{(i,j) \subseteq [n] \times [d] \mid [\nabla f_i(x)]_j = 0 \; \forall x \in \mathbb{R}^d\right\},$$

or, equivalently, $\mathcal{Z} \subseteq [n] \times [d]$ represents the set of pairs $(i,j)$, where $f_i(x)$ does not depend on $x_j$. For instance, if all functions depend on all variables, then $\mathcal{Z} = \emptyset$. Further, we define:

$$\mathcal{I}_j \overset{\text{def}}{=} \{i \in [n] \; : \; (i,j) \notin \mathcal{Z}\}, \qquad \mathcal{J}_i \overset{\text{def}}{=} \{j \in [d] \; : \; (i,j) \notin \mathcal{Z}\}, \tag{7}$$

or, informally, $\mathcal{I}_j$ denotes the set of local loss functions in which the variable $x_j$ is active, and $\mathcal{J}_i$ denotes the set of active variables in the function $f_i$. It is reasonable to assume that $1 \leq |\mathcal{I}_j|$ and $1 \leq |\mathcal{J}_i|$ for all $j \in [d]$ and $i \in [n]$. Otherwise, if $\mathcal{I}_{j'} = 0$, the specific variable $j'$ can be

safely ignored in the equation (1). Similarly, if $\mathcal{I}_{i'} = 0$, we can safely exclude the client $i'$ from consideration as they play no role in (1). Since the union of sets $\mathcal{I}_j$ represents the set of all active variables, which we can express as $[n] \times [d] \setminus \mathcal{Z}$, we write $\sum_{j=1}^{d} |\mathcal{I}_j| = nd - |\mathcal{Z}|$. Similarly, $\sum_{i=1}^{n} |\mathcal{J}_i| = nd - |\mathcal{Z}|$. Hence, we have

$$\frac{1}{n}\frac{1}{d}\sum_{j=1}^{d}|\mathcal{I}_j| = \frac{1}{d}\frac{1}{n}\sum_{i=1}^{n}|\mathcal{J}_i| = 1 - \frac{|\mathcal{Z}|}{nd}. \tag{8}$$

We now define two key sparsity parameters as follows:

$$c \stackrel{\text{def}}{=} \max_{j\in[d]}|\mathcal{I}_j|, \qquad r \stackrel{\text{def}}{=} \max_{i\in[n]}|\mathcal{J}_i|. \tag{9}$$

The parameter $c$ represents the maximum number of clients possessing any variable, while $r$ denotes the maximum number of variables possessed by any client. Therefore, $1 \le c \le n$ and $1 \le r \le d$. Since both $c$ and $r$ represent maximum values in their sets, they are both lower bounded by average values over those sets. The following inequalities result from both observations:

$$1 \ge \frac{c}{n} \ge \frac{1}{n}\frac{1}{d}\sum_{j=1}^{d}|\mathcal{I}_j| \stackrel{(8)}{=} 1 - \frac{|\mathcal{Z}|}{nd}, \qquad 1 \ge \frac{r}{d} \ge \frac{1}{d}\frac{1}{n}\sum_{i=1}^{n}|\mathcal{J}_i| \stackrel{(8)}{=} 1 - \frac{|\mathcal{Z}|}{nd}.$$

An illustrative instance of loss functions having small $c$ and $r$ is represented by generalized linear models with sparse data.

**Example 1** (Linear models with sparse data). *Let $f_i(x) = \ell_i(a_i^\top x)$, $i = 1, 2, \ldots, n$, where $\ell_i : \mathbb{R} \to \mathbb{R}$ are loss functions, and the vectors $a_1, \ldots, a_n \in \mathbb{R}^d$ are sparse. Then*

$$\mathcal{Z} = \{(i,j) \in [n] \times [d] \ : \ \ell_i'(a_i^\top x)a_{ij} = 0, \ \forall x \in \mathbb{R}^d\} \supseteq \{(i,j) \in [n] \times [d] \ : \ a_{ij} = 0\} \stackrel{\text{def}}{=} \mathcal{Z}'.$$

*In this case, $c \le \max_j |\{i \ : \ (i,j) \ne \mathcal{Z}'\}| = \max_j |\{i \ : \ a_{ij} \ne 0\}|$.*

We conclude this section by introducing notation that is essential for our subsequent results:

$$\mathbb{R}_i^d \stackrel{\text{def}}{=} \{u = (u_1, \ldots, u_d) \in \mathbb{R}^d \ : \ u_j = 0 \text{ whenever } (i,j) \in \mathcal{Z}\}. \tag{10}$$

Note that for any $x \in \mathbb{R}^d$, the gradient $\nabla f_i(x)$ belongs to $\mathbb{R}_i^d$.

# 5 THEORY

In this section, we present fundamental insights into how the convergence of EF21 is affected by $c$ and $r$. To accomplish this, we revisit all the crucial elements of the original analysis of EF21 (Richtárik et al., 2021) and enhance it.

## 5.1 AVERAGE SMOOTHNESS

The convergence rate of EF21 is directly affected by Assumption 2 only when the analysis estimates the average smoothness. Specifically, the original analysis uses the inequality

$$\frac{1}{n}\sum_{i=1}^{n}\|\nabla f_i(x) - \nabla f_i(y)\|^2 \stackrel{(6)}{\le} \frac{1}{n}\sum_{i=1}^{n}L_i^2\|x - y\|^2 = \tilde{L}^2\|x - y\|^2,$$

where $\tilde{L}^2 \stackrel{\text{def}}{=} \frac{1}{n}\sum_{i=1}^{n}L_i^2$ (Richtárik et al., 2021). Given that this is the only place in the analysis where the local smoothness constants $L_i$ are relevant, a more intelligent way to analyze Algorithm 1 is to replace this assumption with a less restrictive one presented as follows.

**Assumption 4.** *There exists a constant $L_+ \ge 0$ such that*

$$\frac{1}{n}\sum_{i=1}^{n}\|\nabla f_i(x) - \nabla f_i(y)\|^2 \le L_+^2\|x - y\|^2, \quad \forall x, y \in \mathbb{R}^d. \tag{11}$$

Based on the aforementioned observation, it follows that $L_+^2$ is bounded above by $\tilde{L}^2$. However, we can obtain a more refined upper bound on $L_+^2$ by leveraging the concept of sparsity introduced earlier. Concretely, we present the following lemma which provides a tighter bound on $L_+$.

**Lemma 2.** *If Assumption 2 holds, then Assumption 4 holds with*

$$L_+ \leq \sqrt{\frac{\max_j\left\{\sum_{i:(i,j)\notin \mathcal{Z}} L_i^2\right\}}{n}} \leq \min\left\{\sqrt{\frac{c\max_i L_i^2}{n}}, \tilde{L}\right\}. \tag{12}$$

Let us introduce $L_+(\mathcal{Z}) \stackrel{\text{def}}{=} \sqrt{\frac{\max_j\left\{\sum_{i:(i,j)\notin \mathcal{Z}} L_i^2\right\}}{n}}$. According to Lemma 2, $L_+ \leq L_+(\mathcal{Z})$. Moreover, if $\mathcal{Z}' \supset \mathcal{Z}$, all other factors being equal, then the resulting sparsity level is higher and thus $L_+(\mathcal{Z}') \leq L_+(\mathcal{Z})$. Therefore, we can infer that a higher degree of sparsity leads to a smaller value of $L_+$.

## 5.2 Contraction on $\mathbb{R}_i^d$

Another important insight concerns the contraction parameter of the TopK compressor.

**Lemma 3.** *Consider problem* (1) *and Algorithm 1. Then, for all $i \in [n]$ and $x \in \mathbb{R}_i^d$, we have*

$$\|\mathsf{TopK}_i(x) - x\|^2 \leq \left(1 - \frac{\min\{K_i, |\mathcal{J}_i|\}}{|\mathcal{J}_i|}\right) \|x\|^2. \tag{13}$$

The intuition behind the lemma is that if the active dimension space $|\mathcal{J}_i|$ of the function $f_i$ is less than $d$, then the TopK$_i$ compressor is more efficient. For example, if $K_i = |\mathcal{J}_i|$, then the contractive compressor returns all non-zero components, resulting in the corresponding contraction parameter $\alpha_i = 1$. The lemma implies that if $K_i \equiv K$ for all $i \in [n]$, then the worst contraction parameter is $K/r$.

## 5.3 Interpolating between orthogonality and parallelism

The sparsity parameter $c$ has an important role in interpolating between orthogonal and parallel vectors, as demonstrated in the following lemma.

**Lemma 4.** *If $u_1 \in \mathbb{R}_1^d, \ldots, u_n \in \mathbb{R}_n^d$, then $\left\|\sum_{i=1}^n u_i\right\|^2 \leq c \sum_{i=1}^n \|u_i\|^2$.*

If $c = 1$, then each client $i$ owns a unique set of variables, and hence the scalar products $\langle u_i, u_j \rangle$ are zero for $i \neq j$, since non-zero elements of one vector are multiplied by zeros of another in the product sum. Note that for orthogonal vectors, Lemma 4 becomes an equality with $c = 1$. When $c = n$, Lemma 4 reduces to Young's inequality, which states that for any $u_i \in \mathbb{R}^d$, $\|\sum_{i=1}^n u_i\|^2 \leq n\sum \|u_i\|^2$. This inequality is an equality when all vectors are parallel and of the same length.

In the context of EF21 theory, Lemma 4 provides another enhancement. To present it, we introduce the following quantity representing the error in our estimation of the gradients:

$$G^t \stackrel{\text{def}}{=} \frac{1}{n}\sum_{i=1}^n G_i^t, \qquad G_i^t \stackrel{\text{def}}{=} \|g_i^t - \nabla f_i(x^t)\|^2. \tag{14}$$

The following lemma represents this enhancement.

**Lemma 5.** *Assume that $g_i^0 \in \mathbb{R}_i^d$ for all $i \in [n]$. Then, it holds for all $t \geq 0$ that*

$$\|g^t - \nabla f(x^t)\|^2 \leq \frac{c}{n}G^t. \tag{15}$$

In contrast to the standard analysis that uses the inequality $\|g^t - \nabla f(x^t)\|^2 \leq G^t$, Lemma 5 provides an enhanced result by incorporating the sparsity parameter $c$.

## 5.4 Main theorem

Drawing on the key observations made above, we can now present our main result.

**Theorem 2.** *Let Assumptions 1, 3, 4 hold. Let $\alpha \stackrel{\text{def}}{=} \min_i \alpha_i$, where $\alpha_i \stackrel{\text{def}}{=} \frac{\min\{K_i, |\mathcal{J}_i|\}}{|\mathcal{J}_i|}$, $\theta \stackrel{\text{def}}{=} 1 - \sqrt{1-\alpha}$ and $\beta \stackrel{\text{def}}{=} \frac{1-\alpha}{1-\sqrt{1-\alpha}}$. Choose $\gamma \leq \frac{1}{L+L_+\sqrt{\frac{\beta c}{\theta n}}}$. Under these conditions, the iterates of Algorithm 1 (EF21) satisfy*

$$\frac{1}{T}\sum_{t=0}^{T-1} \|\nabla f(x^t)\|^2 \leq \frac{2\left(f(x^0) - f^*\right)}{\gamma T} + \frac{c}{n}\frac{G^0}{\theta T}. \tag{16}$$

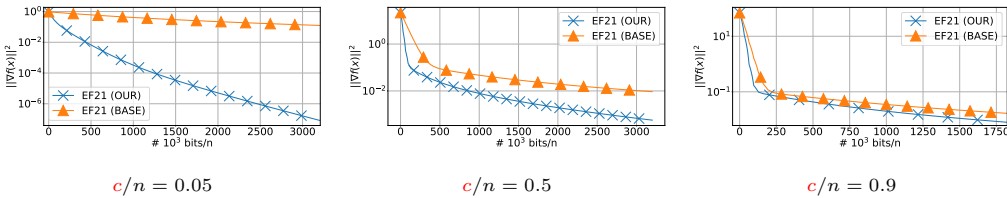

$$c/n = 0.05 \qquad\qquad c/n = 0.5 \qquad\qquad c/n = 0.9$$

Figure 2: Comparison of the performance of EF21 + Top1 with the standard approach proposed by Richtárik et al. (2021) and a newly proposed stepsize (see Theorem 2) on the linear regression problem. The sparsity pattern $c$ is controlled by manipulating the sparsity of the data matrix.

We make the following immediate observations:

1. If we let $K_i = 1$ for all $i \in [n]$, then $\alpha_i = \frac{1}{|\mathcal{J}_i|}$ and $\alpha = \min_i \alpha_i = \frac{1}{\max_i |\mathcal{J}_i|} \overset{(9)}{=} \frac{1}{r}$. Thus, the term appearing in the stepsize can be bounded as follows: $\sqrt{\beta/\theta} = (\sqrt{1-\alpha}+1-\alpha)/\alpha \leq 2\sqrt{1-\alpha}/\alpha = 2r\sqrt{1 - 1/r} = 2\sqrt{r(r-1)} \leq 2r$, where the proof of the first equality is deferred to the appendix. By using Theorem 2, we find that EF21 with Top1 compressor requires $\mathcal{O}\left((L + rL_+ \sqrt{\frac{c}{n}})\frac{1}{\varepsilon}\right)$ bits to converge to the $\varepsilon$-stationary point. Comparing this with the standard communication complexity of DGD $\mathcal{O}(rL\frac{1}{\varepsilon})$, we can see that EF21 gets asymptotically faster if $L_+\sqrt{\frac{c}{n}} \leq L$. In fact, when $K_i \equiv K$ in Algorithm 1, the optimal value for $K$ is either 1 or $r$, as elucidated in Lemma 6 in the appendix.

2. By using the largest stepsize allowed by the theory, we get the bound

$$\frac{1}{T}\sum_{t=0}^{T-1} \|\nabla f(x^t)\|^2 \leq 2\left(L + \min\left\{\sqrt{\frac{c\max_i L_i^2}{n}}, \sqrt{\frac{\sum_{i=1}^n L_i^2}{n}}\right\}\sqrt{\frac{\beta c}{\theta n}}\right)\frac{\Psi^0}{T}. \tag{17}$$

3. In a pessimistic scenario where $c = n$, the bound (17) recovers the standard EF21 rate:

$$\frac{1}{T}\sum_{t=0}^{T-1} \|\nabla f(x^t)\|^2 \leq 2\left(L + \sqrt{\frac{\sum_{i=1}^n L_i^2}{n}}\sqrt{\frac{\beta}{\theta}}\right)\frac{\Psi^0}{T}.$$

4. In an optimistic scenario where $c = 1$, the bound (17) recovers the EF21 rate in the separable regime, discussed in the appendix:

$$\frac{1}{T}\sum_{t=0}^{T-1} \|\nabla f(x^t)\|^2 \leq 2\left(L + \sqrt{\frac{\max_i L_i^2}{n}}\sqrt{\frac{\beta}{\theta n}}\right)\frac{\Psi^0}{T}.$$

In the next section, we perform computational experiments to validate our theoretical results.

## 6 EXPERIMENTS

The practical superiority of EF21 has been demonstrated in various previous publications (Szlendak et al., 2022; Richtárik et al., 2021). These publications have explicitly indicated that the stepsize of EF21 utilized in the experiments is fine-tuned, and is typically substantially larger than the value suggested by theoretical analysis. Consequently, the objective of our experimental section is not to reassert the dominance of the Error Feedback algorithm, but rather to substantiate our theoretical contentions. In order to do so, it suffices to perform small scale but carefully executed experiments.

### 6.1 LINEAR REGRESSION ON SPARSE DATA

Consider the following optimization problem:

$$\min_{x\in\mathbb{R}^d}\left\{f(x) = \frac{1}{n}\sum_{i=1}^n f_i(x) = \frac{1}{n}\sum_{i=1}^n \frac{1}{m}\|\mathbf{A}_i x - b_i\|^2\right\},$$

where $\mathbf{A}_i \in \mathbb{R}^{m\times d}, b_i \in \mathbb{R}^m$ are the sparse training data and labels. Each $f_i$ is a $L_i$-smooth function, where $L_i = \sup_{x\in\mathbb{R}^d}\|\nabla^2 f_i(x)\| = \left\|\frac{2}{m}\mathbf{A}_i^\top \mathbf{A}_i\right\|$ (Nesterov et al., 2018). We show in the appendix

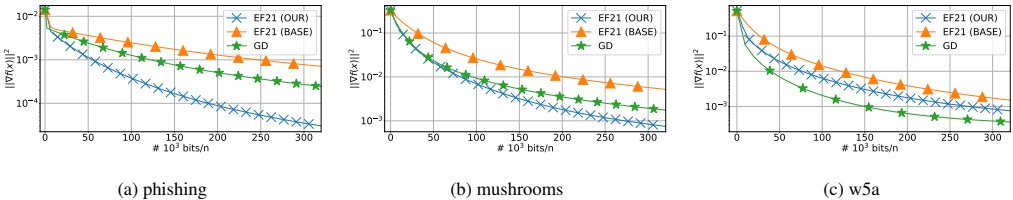

(a) phishing          (b) mushrooms          (c) w5a

Figure 3: Comparison of the performance of the EF21 + Top1 algorithm, with both standard and adaptive stepsize, and the GD method, on a logistic regression problem on the LIBSVM datasets. All stepsizes are theoretical.

that the function $f(x)$ satisfies Assumption 4 with $L_+^2 = \frac{4}{m^2 n} \lambda_{\max} \left( \sum_{i=1}^n (\mathbf{A}_i^\top \mathbf{A}_i)^2 \right)$. In all experiments in this section, we fix $n$, $d$ and $m$ to 500, 100 and 12, respectively.

As the function $f_i$ is a generalized linear function, we can manipulate the sparsity of the data matrix $\mathbf{A}_i$ to adjust the sparsity parameter $c$ (see Example 1). In our experiments, we varied the ratio $c/n$, a critical factor in the theoretical analysis, over the values in the list $[0.05, 0.5, 0.9]$. To strengthen our findings, we controlled individual smoothness constants $L_i$ to ensure that the constant $L_+^2$ from Assumption 4, which is presented in our theoretical results, was much smaller than $\tilde{L}^2 = \frac{1}{n} \sum_{i=1}^n L_i^2$ used by Richtárik et al. (2021). Further details can be found in the appendix.

All experiments were implemented using FL_PyTorch (Burlachenko et al., 2021) and were conducted on two Linux workstations with x86_64 architecture and 48 CPUs each.

The results presented in Figure 2 demonstrate that the performance gap between EF21 with the standard and new stepsizes, as proposed in Theorem 2, is significant when the parameter $c$ is much smaller than its maximum value $n$. Conversely, as $c$ approaches $n$, the difference in performance becomes negligible. This is expected because firstly, the new stepsize scales directly with the ratio $c/n$, and secondly, the constant $L_+^2$, as can be seen from (12), approaches $\tilde{L}^2 = \frac{1}{n} \sum_{i=1}^n L_i^2$ which is utilized in the standard theory. The findings align with the paper's main claim that EF21 achieves faster convergence with smaller sparsity parameter $c$.

## 6.2 LOGISTIC REGRESSION WITH ADAPTIVE STEPSIZE

The inequalities in (12) and (15) rely heavily on the parameter $c$. To explore the possibility of eliminating the sparsity condition altogether, we consider replacing the initial definition of $c$ as a sparsity pattern with (15). In this experimental section, we investigate this question.

We consider the following optimization problem:

$$\min_{x \in \mathbb{R}^d} \left\{ f(x) = \frac{1}{N} \sum_{i=1}^N \log(1 + e^{-y_i a_i^\top x}) \right\},$$

where $a_i \in \mathbb{R}^d, y \in -1, 1$ represent the training data and labels, respectively. We utilize three LIBSVM (Chang and Lin, 2011) datasets, namely *phishing, mushrooms, w5a*, as the training data, dividing the data evenly between $n = 300$ clients.

We compute the parameter $c^t$ according to (15), i.e., $c^t \overset{\text{def}}{=} \frac{n \|g^t - \nabla f(x^t)\|^2}{G^t}$, at each iteration of EF21. We further estimate $L_+^t \overset{\text{def}}{=} \min\{\sqrt{\frac{c^t \max_i L_i^2}{n}}, \sqrt{\frac{\sum_{i=1}^n L_i^2}{n}}\}$, and finally, compute the stepsize using the formula provided in Theorem 2: $\gamma^t \overset{\text{def}}{=} (L + L_+^t \sqrt{\frac{c^t}{n}} \frac{\sqrt{1-\alpha}+1-\alpha}{\alpha})^{-1}$. We admit that computing $c^t$ in its current form is infeasible as it requires the computation of $\nabla f(x^t)$, which is usually unavailable at the master.

As can be seen from Figure 3, our results demonstrate that adaptive computation of $c^t$ can be a promising direction for further investigation.

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

CONTENTS

# A AUXILIARY RESULTS AND MISSING PROOFS

## A.1 PROOF OF LEMMA 1

*Proof.* Let us now recall that EF21 needs at least[1] $T = \frac{2\Delta_f^0 \left(L + \tilde{L}\left(\frac{1+\sqrt{1-\alpha}}{\alpha} - 1\right)\right)}{\varepsilon}$ iterations to ensure $\mathbb{E}\left[\left\|\nabla f(\hat{x}^T)\right\|^2\right] \leq \varepsilon$, where $\Delta_f^0 = f(x^0) - f^*$, $\tilde{L} = \sqrt{\frac{1}{n}\sum_{i=1}^n L_i^2}$, and $\alpha$ is the contraction parameter of the compressor. In the case of TopK, as was noted earlier, $\alpha = \frac{K}{d}$. To find the optimal $\alpha$, which requires the minimum number of communication, we need to minimize $T \cdot K \cdot n$, since at each iteration of the algorithm each of $n$ clients sends $K$ float numbers. Thus,

$$\min_k \frac{2\Delta_f^0 \left(L + \tilde{L}\left(\frac{1+\sqrt{1-\alpha}}{\alpha} - 1\right)\right)}{\varepsilon} \cdot K \cdot n$$

$$\overset{\alpha = \frac{K}{d}}{\Longleftrightarrow} \min_\alpha \left\{\xi(\alpha) \overset{\text{def}}{=} \left(L + \tilde{L}\left(\frac{1+\sqrt{1-\alpha}}{\alpha} - 1\right)\right) \cdot \alpha\right\}.$$

Taking the derivative of the object over $\alpha$, we get

$$
\begin{aligned}
\xi'(\alpha) &= L + \tilde{L}\left(\frac{1+\sqrt{1-\alpha}}{\alpha} - 1\right) + \tilde{L}\alpha\left(-\frac{1}{(1-\sqrt{1-\alpha})^2}\frac{1}{2\sqrt{1-\alpha}}\right) \\
&= L + \tilde{L}\left(\frac{1+\sqrt{1-\alpha}}{\alpha} - 1\right) - \tilde{L}\frac{1+\sqrt{1+\alpha}}{1-\sqrt{1-\alpha}}\frac{1}{2\sqrt{1-\alpha}} \\
&= L - \tilde{L} + \frac{\tilde{L}}{1-\sqrt{1-\alpha}}\left(1 - \frac{1+\sqrt{1+\alpha}}{2\sqrt{1-\alpha}}\right) \\
&= L - \tilde{L} + \frac{\tilde{L}}{(1-\sqrt{1-\alpha})2\sqrt{1-\alpha}}\underbrace{(\sqrt{1-\alpha}-1)}_{\leq 0} \\
&\leq L - \tilde{L} \\
&\leq 0,
\end{aligned}
$$

where the last inequality holds since $L \leq \tilde{L}$. Indeed, $L \leq \frac{1}{n}\sum_{i=1}^n L_i \leq \sqrt{\frac{1}{n}\sum_{i=1}^n L_i^2} = \tilde{L}$. As a result, it can be concluded that the minimum of the objective function is achieved when the value of $\alpha$ is the largest, i.e., $\alpha = 1$). For the specific case of the TopK compressor, choosing $\alpha = 1$ corresponds to selecting $k = d$ in the compressor definition. $\square$

## A.2 OPTIMAL CONTRACTION PARAMETER FOR SPARSE FEATURES

In this subsection, we discuss what the optimal contraction parameter is when sparse features are present. Here, we leverage the central convergence theorem presented in the paper, namely, Theorem 2.

**Lemma 6.** *In Algorithm 1, assume $K_i \equiv K$ and $g_i^0 = \nabla f_i(x^0)$ for all $i \in [n]$. Let us denote $D \overset{\text{def}}{=} \left(L - L_+ + \sqrt{\frac{c}{n}}\right)\left(1 - \frac{1}{r}\right) - L_+\sqrt{\frac{c}{n}}\sqrt{1 - \frac{1}{r}}$. Assume the conditions of Theorem 2 hold.*

*Then, if $D \geq 0$, then the communication complexity of Algorithm 1 is minimized when $K = 1$. Conversely, if $D < 0$, the optimal choice of $K$ is $r$.*

*Proof.* Given the assumption that $g_i^0 = \nabla f_i(x^0)$ for all $i \in [n]$, and in accordance with Theorem 2, the algorithm EF21 requires a minimum of $T = \frac{2(f(x^0)-f^*)\left(L+L_+\sqrt{\frac{c}{n}}\left(\frac{1+\sqrt{1-\alpha}}{\alpha}-1\right)\right)}{\varepsilon}$ iterations to ensure $\frac{1}{T}\sum_{t=0}^{T-1}\|\nabla f(x^t)\|^2 \leq \varepsilon$. In the same way, as presented in Lemma 1, the communication

---

[1] We suppose the initial gradient estimate $g_0^i$ equals $\nabla f_i(x^0)$ as a part of preprocessing step. This zeroes out the second term of Theorem 2 in (Richtárik et al., 2021).

complexity of the algorithm can be expressed as the product of the number of iterations and the cost associated with each iteration. Our objective is to solve the optimization problem defined as:

$$\frac{2(f(x^0) - f^*)\left(L + L_+\sqrt{\frac{c}{n}}\left(\frac{1+\sqrt{1-\alpha}}{\alpha} - 1\right)\right)}{\varepsilon} \cdot K \cdot n$$

$$\overset{\alpha = \frac{K}{\tau}}{\Longleftrightarrow} \min_\alpha \left\{ \xi(\alpha) \overset{\text{def}}{=} \left(L + L_+\sqrt{\frac{c}{n}}\left(\frac{1+\sqrt{1-\alpha}}{\alpha} - 1\right)\right)\right\} \cdot \alpha.$$

To simplify the expression, let us introduce the notation $B \overset{\text{def}}{=} L_+\sqrt{\frac{c}{n}}$. Thus, we have $\xi(\alpha) = L\alpha + B(1 - \alpha + \sqrt{1-\alpha})$. The derivative $\xi'(\alpha)$ is given by $L - B - \frac{B}{2\sqrt{1-\alpha}}$. Notably, the second derivative $\xi''(\alpha) = -\frac{B}{4}(1-\alpha)^{-\frac{3}{2}}$ is negative over the entire interval, implying that the function $\xi(\alpha)$ is *concave*. Consequently, the minimum is achieved at one of the endpoints of the interval $\left[\frac{1}{r}, 1\right]$.

We observe that $\xi(1) = L$, and $\xi\left(\frac{1}{r}\right) = \frac{L}{r} + B\left(1 - \frac{1}{r} + \sqrt{1 - \frac{1}{r}}\right)$. The difference between these values is $\xi(1) - \xi\left(\frac{1}{r}\right) = (L - B)\left(1 - \frac{1}{r}\right) - B\sqrt{1 - \frac{1}{r}} = D$. Hence, if $\xi(1) - \xi\left(\frac{1}{d}\right) \geq 0$, then the minimum communication complexity is attained at the minimum $\alpha$, and if it is negative, then the optimal parameter $\alpha$ equals 1. $\qquad\square$

### A.3 REFINING SMOOTHNESS WHEN SPARSITY IS PRESENT

**Lemma 7.** *If Assumption 2 holds, then for every $i \in [n]$, we have*

$$\sum_{j:(i,j)\notin \mathcal{Z}} ((\nabla f_i(x))_j - (\nabla f_i(y))_j)^2 \leq L_i^2 \sum_{j:(i,j)\notin \mathcal{Z}} (x_j - y_j)^2, \qquad \forall x, y \in \mathbb{R}^d. \tag{18}$$

*Proof.* Given any $i \in [n]$ and any $x, y \in \mathbb{R}^d$, we know that

$$\sum_{j:(i,j)\notin \mathcal{Z}} ((\nabla f_i(x))_j - (\nabla f_i(y))_j)^2 = \sum_{j=1}^{d} ((\nabla f_i(x))_j - (\nabla f_i(y))_j)^2$$

$$= \|\nabla f_i(x) - \nabla f_i(y)\|^2$$

$$\leq L_i^2 \|x - y\|^2, \tag{19}$$

where the last inequality follows from Assumption 2. Let $x'$ and $y'$ be formed from $x$ and $y$ by replacing the coordinates $j$ for which $(i,j) \in \mathcal{Z}$ by zeros. That is,

$$x'_j = \begin{cases} x_j & (i,j) \notin \mathcal{Z} \\ 0 & (i,j) \in \mathcal{Z} \end{cases}, \qquad y'_j = \begin{cases} y_j & (i,j) \notin \mathcal{Z} \\ 0 & (i,j) \in \mathcal{Z} \end{cases}. \tag{20}$$

Applying inequality (19) with $x \leftarrow x'$ and $y \leftarrow y'$, we obtain

$$\sum_{j:(i,j)\notin \mathcal{Z}} ((\nabla f_i(x'))_j - (\nabla f_i(y'))_j)^2 \leq L_i^2 \|x' - y'\|^2$$

$$= L_i^2 \left( \sum_{j:(i,j)\notin \mathcal{Z}} (x'_j - y'_j)^2 + \sum_{j:(i,j)\in \mathcal{Z}} (x'_j - y'_j)^2 \right)$$

$$\overset{(20)}{=} L_i^2 \left( \sum_{j:(i,j)\notin \mathcal{Z}} (x_j - y_j)^2 + \sum_{j:(i,j)\in \mathcal{Z}} (0 - 0)^2 \right)$$

$$= L_i^2 \sum_{j:(i,j)\notin \mathcal{Z}} (x_j - y_j)^2. \tag{21}$$

The result now follows by comparing the left-hand side of (19) and the right-hand side of (21) in view of the observation that $\nabla f_i(x) = \nabla f_i(x')$ and $\nabla f_i(y) = \nabla f_i(y')$. $\qquad\square$

## A.4 PROOF OF LEMMA 2

*Proof.* Using Lemma 7, we can now write

$$
\frac{1}{n}\sum_{i=1}^{n}\|\nabla f_i(x) - \nabla f_i(y)\|^2 = \frac{1}{n}\sum_{i=1}^{n}\sum_{j=1}^{d}((\nabla f_i(x))_j - (\nabla f_i(y))_j)^2
$$

$$
= \frac{1}{n}\sum_{i=1}^{n}\sum_{j:(i,j)\notin\mathcal{Z}}((\nabla f_i(x))_j - (\nabla f_i(y))_j)^2
$$

$$
\overset{(18)}{\leq} \frac{1}{n}\sum_{i=1}^{n}L_i^2\sum_{j:(i,j)\notin\mathcal{Z}}(x_j - y_j)^2 \tag{22}
$$

$$
= \frac{1}{n}\sum_{i=1}^{n}\sum_{j:(i,j)\notin\mathcal{Z}}L_i^2(x_j - y_j)^2
$$

$$
= \frac{1}{n}\sum_{j=1}^{d}\sum_{i:(i,j)\notin\mathcal{Z}}L_i^2(x_j - y_j)^2
$$

$$
= \frac{1}{n}\sum_{j=1}^{d}\left[(x_j - y_j)^2\sum_{i:(i,j)\notin\mathcal{Z}}L_i^2\right].
$$

To further advance our analysis, we proceed by determining the maximum value in each individual sum term.

$$
\frac{1}{n}\sum_{i=1}^{n}\|\nabla f_i(x) - \nabla f_i(y)\|^2 \overset{(22)}{\leq} \frac{1}{n}\sum_{j=1}^{d}\left[(x_j - y_j)^2\sum_{i:(i,j)\notin\mathcal{Z}}L_i^2\right]
$$

$$
\leq \frac{1}{n}\sum_{j=1}^{d}\left[(x_j - y_j)^2\max_{j}\left\{\sum_{i:(i,j)\notin\mathcal{Z}}L_i^2\right\}\right]
$$

$$
= \frac{\max_{j}\left\{\sum_{i:(i,j)\notin\mathcal{Z}}L_i^2\right\}}{n}\sum_{j=1}^{d}(x_j - y_j)^2
$$

$$
= \frac{\max_{j}\left\{\sum_{i:(i,j)\notin\mathcal{Z}}L_i^2\right\}}{n}\|x - y\|^2.
$$

Based on the preceding inequality, it can be deduced that $L_+^2 \leq \frac{\max_{j}\left\{\sum_{i:(i,j)\notin\mathcal{Z}}L_i^2\right\}}{n}$. Furthermore,

$$
\frac{\max_{j}\left\{\sum_{i:(i,j)\notin\mathcal{Z}}L_i^2\right\}}{n} \leq \frac{\max_{j}\left\{\sum_{i:(i,j)\notin\mathcal{Z}}\max_i L_i^2\right\}}{n}
$$

$$
= \frac{\left(\max_i L_i^2\right)\left\{\max_j\sum_{i:(i,j)\notin\mathcal{Z}}1\right\}}{n} \overset{(9)}{=} \frac{\left(\max_i L_i^2\right)c}{n},
$$

and

$$
\frac{\max_{j}\left\{\sum_{i:(i,j)\notin\mathcal{Z}}L_i^2\right\}}{n} \leq \frac{\max_{j}\left\{\sum_{i=1}^{n}L_i^2\right\}}{n} = \frac{\sum_{i=1}^{n}L_i^2}{n}.
$$

This concludes the proof of the lemma. $\qquad\square$

## A.5 PROOF OF LEMMA 3

*Proof.* Choose $i \in [n]$ and $x \in \mathbb{R}_i^d$. If $|\mathcal{J}_i| = d$, the statement turns into the standard contraction property of $\mathsf{TopK_i}$ on $\mathbb{R}^d$, and hence it holds[2]. Assume therefore that $|\mathcal{J}_i| < d$. If $x = 0$, inequality

---

[2]The standard contraction property says that $\|\mathsf{TopK_i}(x) - x\|^2 \leq \left(1 - \frac{K_i}{d}\right)\|x\|^2$, for all $x \in \mathbb{R}^d$.

(13) clearly holds. Therefore, let us assume that $x \neq 0$. Notice that $\operatorname{supp}(x) \stackrel{\text{def}}{=} \{j \in [d] \; : \; x_j \neq 0\} \subseteq \mathcal{J}_i$. Hence,

$$s \stackrel{\text{def}}{=} |\operatorname{supp}(x)| \leq |\mathcal{J}_i| < d.$$

Since neither the left nor the right hand side of (13) changes if we permute the coordinates of $x$, we can w.l.o.g. assume that $|x_1| \geq |x_2| \geq \cdots \geq |x_d|$. Notice that $|x_s| > 0$ and that $|x_{s+1}| = \cdots = |x_d| = 0$. Let $y = \mathsf{TopK}_i(x)$, and notice that $y_j = x_j$ for $1 \leq j \leq K_i$, $y_i = 0$ for $j > K_i$, and $y_j = x_j = 0$ for $s + 1 \leq j \leq d$. If $K_i \geq s$, then $y_j = x_j$ for all $j$, which means that the left-hand side in (13) is equal to zero. Therefore, inequality (13) holds. If $K_i < s$, then

$$
\begin{aligned}
\|y - x\|^2 &= \sum_{j=1}^{K_i} (y_j - x_j)^2 + \sum_{j=K_i+1}^{s} (y_j - x_j)^2 + \sum_{j=s+1}^{d} (y_j - x_j)^2 \\
&= \sum_{j=1}^{K_i} (x_j - x_j)^2 + \sum_{j=K_i+1}^{s} (0 - x_j)^2 + \sum_{j=s+1}^{d} (0 - 0)^2 \\
&= \sum_{j=K_i+1}^{s} x_j^2.
\end{aligned}
\tag{23}
$$

Because $x_1^2, x_2^2, \ldots, x_d^2$ is non-increasing, we have

$$\frac{1}{s} \sum_{j=1}^{s} x_j^2 \geq \frac{1}{s - K_i} \sum_{j=K_i+1}^{s} x_j^2.$$

Plugging this estimate into (23), we get

$$\|y - x\|^2 \leq \frac{s - K_i}{s} \sum_{j=1}^{s} x_j^2 = \left(1 - \frac{K_i}{s}\right) \|x\|^2.$$

It remains to apply the bound $s \leq |\mathcal{J}_i|$ and use the identity $K_i = \min\{K_i, s\} = \min\{K_i, |\mathcal{J}_i|\}$. $\quad\square$

### A.6 HELPING LEMMA ON ORTHOGONALITY

**Lemma 8.** *Let $u_1, \ldots, u_n \in \mathbb{R}^d$. Let us write $u_i = (u_{i1}, \ldots, u_{id}) \in \mathbb{R}^d$, and define sets $S_j \stackrel{\text{def}}{=} \{i \; : \; u_{ij} \neq 0\}$ for $j = 1, 2, \ldots, d$. Then*

$$\left\|\sum_{i=1}^{n} u_i\right\|^2 \leq \left(\max_j |S_j|\right) \times \sum_{i=1}^{n} \|u_i\|^2.
\tag{24}$$

*Proof.* First, we rewrite the left-hand side of (24) into the form

$$\left\|\sum_{i=1}^{n} u_i\right\|^2 = \sum_{j=1}^{d} \left(\sum_{i=1}^{n} u_{ij}\right)^2 = \sum_{j=1}^{d} \left(\sum_{i \in S_j} u_{ij}\right)^2.
\tag{25}$$

Let $\xi \stackrel{\text{def}}{=} \max_j |S_j|$. Notice that by Jensen's inequality, we have

$$\left(\sum_{i \in S_j} u_{ij}\right)^2 \leq |S_j| \sum_{i \in S_j} u_{ij}^2 \leq \xi \sum_{i \in S_j} u_{ij}^2
\tag{26}$$

for all $j \in [d]$. By combining the above observations, we can continue as follows:

$$\left\|\sum_{i=1}^{n} u_i\right\|^2 \stackrel{(25)+(26)}{\leq} \xi \sum_{j=1}^{d} \sum_{i \in S_j} u_{ij}^2 = \xi \sum_{j=1}^{d} \sum_{i=1}^{n} u_{ij}^2 = \xi \sum_{i=1}^{n} \sum_{j=1}^{d} u_{ij}^2 = \xi \sum_{i=1}^{n} \|u_i\|^2.$$

$\square$

### A.7 PROOF OF LEMMA 4

*Proof.* Let $S_j \overset{\text{def}}{=} \{i \; : \; u_{ij} \neq 0\}$ for $j \in [d]$. Since

$$S_j = \{i \; : \; u_{ij} \neq 0\} = \{i \; : \; u_{ij} = 0\}^c \subseteq \{i \; : \; (i,j) \in \mathcal{Z}\}^c = \{i \; : \; (i,j) \notin \mathcal{Z}\},$$

we have

$$\max_j |S_j| \leq \max_j |\{i \; : \; (i,j) \notin \mathcal{Z}\}| \overset{(9)}{=} c.$$

It remains to apply Lemma 8. $\qquad\square$

### A.8 GRADIENT ESTIMATE $g_i$ STAYS WITHIN ITS ACTIVE SUBSPACE

**Lemma 9.** *Choose any $i \in [n]$, $K_i \in [d]$ and $x \in \mathbb{R}^d$. If $g_i \in \mathbb{R}^d_i$, then the vector*

$$g_i^+ \overset{\text{def}}{=} g_i + \mathsf{TopK_i}(\nabla f_i(x) - g_i) \tag{27}$$

*also belongs to $\mathbb{R}^d_i$.*

*Proof.* Let us note that the complement set of $\mathcal{J}_i$, as indicated by the definition of $\mathcal{J}_i$ in Equation (7), is given by

$$\mathcal{J}_i^{\complement} = \{j \in [d] \; : \; (i,j) \in \mathcal{Z}\}.$$

We define $l_i = |\mathcal{J}_i^{\complement}|$, which represents the cardinality of $\mathcal{J}_i^{\complement}$. It is now observed that the lemma's statement is equivalent to demonstrating that $[g_i^+]_{\mathcal{J}_i^{\complement}} = \mathbf{0}^{l_i}$. In order to prove this, we further note that

$$[g_i^+]_{\mathcal{J}_i^{\complement}} \overset{(27)}{=} [g_i + \mathsf{TopK_i}(\nabla f_i(x) - g_i)]_{\mathcal{J}_i^{\complement}} = [g_i]_{\mathcal{J}_i^{\complement}} + [\mathsf{TopK_i}(\nabla f_i(x) - g_i)]_{\mathcal{J}_i^{\complement}}, \tag{28}$$

where the last equality follows from basic arithmetic principles. Since $g_i \in \mathbb{R}^d_i$, it holds that $[g_i]_{\mathcal{J}_i^{\complement}} = \mathbf{0}^{l_i}$. Regarding the argument of $\mathsf{TopK_i}$, we can express it as

$$[\nabla f_i(x) - g_i]_{\mathcal{J}_i^{\complement}} = [\nabla f_i(x)]_{\mathcal{J}_i^{\complement}} - [g_i]_{\mathcal{J}_i^{\complement}} = \mathbf{0}^{l_i} - \mathbf{0}^{l_i} = \mathbf{0}^{l_i},$$

since $\nabla f_i(x) \in \mathbb{R}^d_i$ for any $x \in \mathbb{R}^d$. It remains to recall that the $\mathsf{TopK_i}$ operator either retains the element of a vector or maps it to zero. Thus, the zero sub-vector $[\nabla f_i(x) - g_i]_{\mathcal{J}_i^{\complement}}$ is mapped to a zero sub-vector. Consequently,

$$[g_i^+]_{\mathcal{J}_i^{\complement}} \overset{(28)}{=} [g_i]_{\mathcal{J}_i^{\complement}} + [\mathsf{TopK_i}(\nabla f_i(x) - g_i)]_{\mathcal{J}_i^{\complement}} = \mathbf{0}^{l_i} + \mathbf{0}^{l_i} = \mathbf{0}^{l_i},$$

what concludes the proof. $\qquad\square$

### A.9 PROOF OF LEMMA 5

*Proof.* By assumption, $g_i^0 \in \mathbb{R}^d_i$. By repeatedly applying Lemma 9, we conclude that $g_i^t \in \mathbb{R}^d_i$ for all $t \geq 0$. Since $\nabla f_i(x^t)$ belongs to $\mathbb{R}^d_i$, so does the vector $u_i^t \overset{\text{def}}{=} g_i^t - \nabla f_i(x^t)$. We now have

$$\left\| g^t - \nabla f(x^t) \right\|^2 = \left\| \frac{1}{n} \sum_{i=1}^{n} \left( g_i^t - \nabla f_i(x^t) \right) \right\|^2 = \frac{1}{n^2} \left\| \sum_{i=1}^{n} u_i^t \right\|^2 \leq \frac{c}{n^2} \sum_{i=1}^{n} \left\| u_i^t \right\|^2,$$

where in the last step we applied Lemma 4. $\qquad\square$

### A.10 NEW DESCENT LEMMA

**Lemma 10.** *Let Assumption 1 hold. Furthermore, let $g_i^0 \in \mathbb{R}^d_i$ for all $i = 1, 2, \ldots, n$. Let*

$$x^{t+1} = x^t - \gamma g^t$$

*be the* $\mathsf{EF21}$ *method, where $g^t = \frac{1}{n} \sum_{i=1}^{n} g_i^t$, and $\gamma > 0$ is the stepsize. Then*

$$f(x^{t+1}) \leq f(x^t) - \frac{\gamma}{2} \left\| \nabla f(x^t) \right\|^2 - \left( \frac{1}{2\gamma} - \frac{L}{2} \right) \left\| x^{t+1} - x^t \right\|^2 + \frac{\gamma}{2} \frac{c}{n} G^t. \tag{29}$$

We start with a standard result (Li et al., 2021).

**Fact 1.** *Suppose Assumption 1 holds, and let $x^{t+1} = x^t - \gamma g^t$, where $g^t \in \mathbb{R}^d$ is any vector, and $\gamma > 0$ any scalar. Then*

$$f(x^{t+1}) \leq f(x^t) - \frac{\gamma}{2} \left\| \nabla f(x^t) \right\|^2 - \left( \frac{1}{2\gamma} - \frac{L}{2} \right) \left\| x^{t+1} - x^t \right\|^2 + \frac{\gamma}{2} \left\| g^t - \nabla f(x^t) \right\|^2. \quad (30)$$

*Proof.* Lemma 10 follows by plugging the inequality from Lemma 5 into the inequality described by Fact 1. $\qquad \square$

### A.11 BOUNDING THE GRADIENT ESTIMATE ERROR

**Lemma 11.** *The iterates of the EF21 method satisfy*

$$G_i^{t+1} \leq (1-\theta)G_i^t + \beta \left\| \nabla f_i(x^{t+1}) - \nabla f_i(x^t) \right\|^2, \quad (31)$$

*and*

$$G^{t+1} \leq (1-\theta)G^t + \beta L_+^2 \left\| x^{t+1} - x^t \right\|^2, \quad (32)$$

*where $\theta \stackrel{def}{=} 1 - \sqrt{1-\alpha}$, $\beta \stackrel{def}{=} \frac{1-\alpha}{1-\sqrt{1-\alpha}}$, $\alpha \stackrel{def}{=} \min_i \alpha_i$ and $\alpha_i \stackrel{def}{=} \frac{\min\{K_i, |\mathcal{J}_i|\}}{|\mathcal{J}_i|}$.*

*Proof.*

$$
\begin{aligned}
G_i^{t+1} &\stackrel{(14)}{=} \left\| g_i^{t+1} - \nabla f_i(x^{t+1}) \right\|^2 \\
&\stackrel{\text{Step 7 of Alg 1}}{=} \left\| g_i^t + \mathsf{TopK_i}(\nabla f_i(x^{t+1}) - g_i^t) - \nabla f_i(x^{t+1}) \right\|^2 \\
&= \left\| \mathsf{TopK_i}(\nabla f_i(x^{t+1}) - g_i^t) - (\nabla f_i(x^{t+1}) - g_i^t) \right\|^2 \\
&\stackrel{(13)}{\leq} (1-\alpha_i) \left\| \nabla f_i(x^{t+1}) - g_i^t \right\|^2 \\
&\leq (1-\alpha) \left\| \nabla f_i(x^{t+1}) - g_i^t \right\|^2 \\
&= (1-\alpha) \left\| \nabla f_i(x^t) - g_i^t + \nabla f_i(x^{t+1}) - \nabla f_i(x^t) \right\|^2 \\
&\leq (1-\alpha)(1+\zeta) \left\| \nabla f_i(x^t) - g_i^t \right\|^2 + (1-\alpha)(1+\zeta^{-1}) \left\| \nabla f_i(x^{t+1}) - \nabla f_i(x^t) \right\|^2,
\end{aligned}
$$

where $\zeta > 0$ is arbitrary. By choosing $\zeta = \frac{1}{\sqrt{1-\alpha}} - 1$, we obtain (31). To establish (32), we only need to observe that

$$
\begin{aligned}
G^{t+1} &\stackrel{(14)}{=} \frac{1}{n} \sum_{i=1}^{n} G_i^{t+1} \\
&\stackrel{(31)}{\leq} \frac{1}{n} \sum_{i=1}^{n} \left( (1-\theta)G_i^t + \beta \left\| \nabla f_i(x^{t+1}) - \nabla f_i(x^t) \right\|^2 \right) \\
&= (1-\theta)\frac{1}{n} \sum_{i=1}^{n} G_i^t + \beta \frac{1}{n} \sum_{i=1}^{n} \left\| \nabla f_i(x^{t+1}) - \nabla f_i(x^t) \right\|^2 \\
&\stackrel{(11)+(14)}{\leq} (1-\theta)G^t + \beta L_+^2 \left\| x^{t+1} - x^t \right\|^2.
\end{aligned}
$$

$\qquad \square$

### A.12 AUXILIARY RESULT ON CONNECTION BETWEEN $\sqrt{\frac{\beta}{\theta}}$ AND $\alpha$

**Lemma 12.** *Let $\theta \stackrel{def}{=} 1 - \sqrt{1-\alpha}$ and $\beta \stackrel{def}{=} \frac{1-\alpha}{1-\sqrt{1-\alpha}}$, where $\alpha \in (0,1]$. Then,*

$$\sqrt{\frac{\beta}{\theta}} = \frac{\sqrt{1-\alpha} + 1 - \alpha}{\alpha}.$$

*Proof.* It immediately holds from the following arithmetical operations:

$$\sqrt{\frac{\beta}{\theta}} = \sqrt{\frac{1-\alpha}{(1-\sqrt{1-\alpha})^2}} = \frac{\sqrt{1-\alpha}}{1-\sqrt{1-\alpha}} = \frac{\sqrt{1-\alpha}(1+\sqrt{1-\alpha})}{(1-\sqrt{1-\alpha})(1+\sqrt{1-\alpha})} = \frac{\sqrt{1-\alpha}+1-\alpha}{\alpha}.$$

$\square$

### A.13 CONVERGENCE RATE IN THE FULLY SEPARABLE CASE

If $c = 1$, we can express the problem (1) in a more concise form:

$$\min_{x \in \mathbb{R}^d} \left[ f(x) = \frac{1}{n} \sum_{i=1}^{n} f_i(x_i) \right], \tag{33}$$

where each function $f_i$ has a support of size $d_i$, and the sum of all $d_i$ values equals $d$. We will now establish the following claim.

**Lemma 13.** *Let Assumption 2 hold. Then Assumption 1 holds for* (33) *with* $L = \frac{\max_i L_i}{n}$.

*Proof.* For the sake of clarity and ease of presentation, we make the assumption that function $f$ is twice differentiable. We observe that the Hessian matrix of $f(x)$ is a block-separable matrix, given by $\nabla^2 f(x) = \frac{1}{n} \sum_{i=1}^{n} \nabla^2 f_i(x_i)$ for any $x \in \mathbb{R}^d$. To establish the main claim of the lemma, we aim to show that $\|\nabla^2 f(x)\| \leq \frac{\max_i L_i}{n}$ (Nesterov et al., 2018).

Let $v \in \mathbb{R}^d$ such that $\|v\|^2 = 1$. We represent the vector $v = [v_1^\top v_2^\top \ldots v_n^\top]^\top$. It follows that

$$v^\top \nabla^2 f(x) v = \frac{1}{n} \sum_{i=1}^{n} v_i^\top \nabla^2 f_i(x_i) v_i,$$

due to the block-separable structure of the Hessian matrix. We proceed to

$$v^\top \nabla^2 f(x) v = \frac{1}{n} \sum_{i=1}^{n} v_i^\top \nabla^2 f_i(x_i) v_i \leq \frac{1}{n} \sum_{i=1}^{n} \|\nabla^2 f_i(x_i)\| \|v_i\|^2 \leq \frac{1}{n} \sum_{i=1}^{n} L_i \|v_i\|^2 = \sum_{i=1}^{n} \|v_i\|^2 \cdot \frac{L_i}{n}, \tag{34}$$

where the first inequality follows from the definition of the operator norm, and the second inequality follows from the second-order definition of the function smoothness (Nesterov et al., 2018).

Since $1 = \|v\|^2 = \sum_{i=1}^{n} \|v_i\|^2$, we can interpret (34) as a convex combination of $n$ positive numbers. As the convex combination never exceeds the value of its maximum term, we finally obtain

$$v^\top \nabla^2 f(x) v \overset{(34)}{\leq} \sum_{i=1}^{n} \|v_i\|^2 \cdot \frac{L_i}{n} \leq \frac{\max_i L_i}{n},$$

which concludes the proof. $\square$

We observe that applying Algorithm 1 to the problem (33) is equivalent to performing $n$ independent runs of Algorithm 1 in a single-node scenario, where each run pertains to its own $i$-th block. Each individual run requires $\mathcal{O}\left(\frac{L_i(1+\sqrt{\frac{\beta}{\theta}})}{\delta}\right)$ iterations to achieve an accuracy of $\delta$ (Richtárik et al., 2021). Consequently, when considering all $n$ runs, the total number of iterations $T'$ required to attain $\delta$-accuracy by each client is given by $T' = \mathcal{O}\left(\frac{\max_i L_i(1+\sqrt{\frac{\beta}{\theta}})}{\delta}\right)$ in a parallel computing setting.

Since the functions $f_i$ are effectively independent of each other, after $T'$ iterations, the function $f(x)$ satisfies

$$\left\|\nabla f(x^{T'})\right\|^2 = \left\|\frac{1}{n} \sum_{i=1}^{n} f_i(x_i^{T'})\right\|^2 = \frac{1}{n^2} \sum_{i=1}^{n} \left\|\nabla f_i(x^{T'})\right\|^2 \leq \frac{\delta}{n} = \varepsilon.$$

Consequently, to achieve an accuracy of $\varepsilon$ for the function $f(x)$, each parallel run should be executed for

$$\mathcal{O}\left(\frac{\max_i L_i(1 + \sqrt{\frac{\beta}{\theta}})}{\varepsilon n}\right) \overset{\text{Lemma 13}}{=} \mathcal{O}\left(\frac{L + \frac{\max_i L_i}{n}\sqrt{\frac{\beta}{\theta}}}{\varepsilon}\right)$$

iterations, which aligns with the convergence result presented in Theorem 2.

## B  PROOF OF THEOREM 2

*Proof.* Define the Lyapunov function

$$\Psi^t \stackrel{\text{def}}{=} f(x^t) - f^* + \frac{\gamma c}{2\theta n} G^t. \tag{35}$$

By straightforward arguments, we get

$$
\begin{aligned}
\Psi^{t+1} \stackrel{(35)}{=} & \ f(x^{t+1}) - f^* + \frac{\gamma}{2\theta n} G^{t+1} \\
\stackrel{(10)}{\leq} & \ f(x^t) - f^* - \frac{\gamma}{2} \left\| \nabla f(x^t) \right\|^2 - \left( \frac{1}{2\gamma} - \frac{L}{2} \right) \left\| x^{t+1} - x^t \right\|^2 + \frac{\gamma c}{2n} G^t + \frac{\gamma c}{2\theta n} G^{t+1} \\
\stackrel{(32)}{\leq} & \ f(x^t) - f^* - \frac{\gamma}{2} \left\| \nabla f(x^t) \right\|^2 - \left( \frac{1}{2\gamma} - \frac{L}{2} \right) \left\| x^{t+1} - x^t \right\|^2 + \frac{\gamma c}{2n} G^t \\
& + \frac{\gamma c}{2\theta n} \left( (1 - \theta) G^t + \beta L_+^2 \left\| x^{t+1} - x^t \right\|^2 \right) \\
= & \ f(x^t) - f^* + \frac{\gamma}{2} \left( \frac{c}{n} + (1 - \theta) \frac{c}{\theta n} \right) G^t - \frac{\gamma}{2} \left\| \nabla f(x^t) \right\|^2 \\
& - \underbrace{\left( \frac{1}{2\gamma} - \frac{L}{2} - \frac{\gamma \beta c L_+^2}{2\theta n} \right)}_{\geq 0} \left\| x^{t+1} - x^t \right\|^2 \\
\leq & \ f(x^t) - f^* + \frac{\gamma}{2} \left( \frac{c}{n} + (1 - \theta) \frac{c}{\theta n} \right) G^t - \frac{\gamma}{2} \left\| \nabla f(x^t) \right\|^2 \\
= & \ f(x^t) - f^* + \frac{\gamma c}{2\theta n} G^t - \frac{\gamma}{2} \left\| \nabla f(x^t) \right\|^2 \\
\stackrel{(35)}{=} & \ \Psi^t - \frac{\gamma}{2} \left\| \nabla f(x^t) \right\|^2.
\end{aligned}
$$

Unrolling the inequality above, we get

$$0 \leq \Psi^T \leq \Psi^{T-1} - \frac{\gamma}{2} \left\| \nabla f(x^{T-1}) \right\|^2 \leq \Psi^0 - \frac{\gamma}{2} \sum_{t=0}^{T-1} \left\| \nabla f(x^t) \right\|^2, \tag{36}$$

and the result follows. $\qquad \square$

## C  ADDITIONAL DETAILS OF EXPERIMENTS

### C.1  LINEAR REGRESSION ON SPARSE DATA

In our synthetic experiments, we consider the minimization of the function

$$f(x) = \frac{1}{n} \sum_{i=1}^{n} f_i(x),$$

where $f_i(x) = \frac{1}{m} \|\mathbf{A}_i x - b_i\|^2 + \phi(x)$ and and we choose $\phi \equiv 0$. Therefore,

$$\nabla f_i(x) = \frac{2}{m} \left( \mathbf{A}_i^\top \mathbf{A}_i x - \mathbf{A}_i^\top b_i \right).$$

**The function $f(x)$ satisfies Assumption 4.**  The statement follows from since for all $x, y \in \mathbb{R}^d$, we have

$$
\begin{aligned}
\frac{1}{n} \sum_{i=1}^{n} \|\nabla f_i(x) - \nabla f_i(y)\|^2 &= \frac{1}{n} \sum_{i=1}^{n} \left\| \frac{2}{m} \mathbf{A}_i^\top \mathbf{A}_i x - \frac{2}{m} \mathbf{A}_i^\top \mathbf{A}_i y \right\|^2 \\
&= \frac{1}{n} \sum_{i=1}^{n} \frac{4}{m^2} \left\| \mathbf{A}_i^\top \mathbf{A}_i (x - y) \right\| \\
&= (x - y)^\top \left[ \frac{4}{m^2 n} \sum_{i=1}^{n} (\mathbf{A}_i^\top \mathbf{A}_i)^2 \right] (x - y) \\
&\leq L_+^2 \|x - y\|^2 .
\end{aligned}
$$

From this we can conclude that for this problem functions $f_1(x), \ldots, f_n(x)$ satisfy Equation (11) with:

$$L_+^2 = \frac{4}{m^2 n} \lambda_{\max} \left( \sum_{i=1}^{n} (\mathbf{A}_i^\top \mathbf{A}_i)^2 \right).$$

In our experiments, we fixed the dimension $d = 500$ and the number of clients $n = 100$. For the analysis, we designed a controlled way to generate instances of synthetic quadratic optimization problems with a desired sparsity pattern. The generation of an instance of optimization problems is driven by the main meta parameter $c/n$, and the auxiliary meta-parameter $v$ which affects the distribution of $L_i$.

**Ensuring $L_+^2 \ll \tilde{L}^2$.**  To fully demonstrate the efficacy of the new theoretical results, it is desirable to enforce a significant difference between $L_+^2$ and $\tilde{L}^2$. The standard theory assumes that $L_+^2 = \tilde{L}^2$ (Richtárik et al., 2021), but the refined Lemma 2 allows for the reduction of $L_+^2$. Assuming that $\max_i L_i^2$ is attained at index $j'$, if we modify $L_i$ for $i \neq j'$ such that $L_i \leq L_{j'}$, the left-hand side of the expression

$$\min \left\{ \sqrt{\frac{c \max_i L_i^2}{n}}, \sqrt{\frac{\sum_{i=1}^{n} L_i^2}{n}} \right\}.$$

remains unaffected, while the second term, equal to $\tilde{L}$, is influenced. Clearly, for a fixed value of $\max_i L_i^2 = L_{j'}^2$, in order to maximize $\tilde{L}$, we need to select $L_i = L_j, \forall i \in [n]$. The instances of a quadratic optimization problem with such a property demonstrate the greatest advantage of the new theory.

On the other hand, if we ask the question when our analysis does not bring a big improvement over the standard EF21 analysis, this is the case when $\max_i L_i^2 = \sum_{i=1}^{n} L_i^2$. This situation is attained when $L_{j'}$ is constant and $L_i = 0$ for $i \neq j'$.

**Main meta parameter $c/n$.** From Lemma 2, we see that $\frac{c}{n}$ plays an important role in the multiplier in Lemma 2 and comes into the denominator of the stepsize in Theorem 2. Firstly, we can observe from Equation (9) that the minimum possible value of $\frac{c}{n} = \frac{1}{n}$, and the maximum possible value of $\frac{c}{n} = 1$ is attained for $c = n$. So $\frac{c}{n} \in \left[\frac{1}{n}, 1\right]$. We provide a controllable way to specify this parameter. The main intuition behind this parameter is the following. As $\frac{c}{n}$ is smaller, it is more advantageous for our method compared to the standard EF21. And in this case, there is a serious hope to observe in practice that our strategy of selecting the step size demonstrates better results

**Auxiliary parameter $v$.** The meta-parameter $v \in [0, 1]$ allows selecting between two extreme distributions of $L_i$ in context of Lemma 2. One extreme point is when all $L_i$ attains the same constant values $L_c$ (can be selected arbitrarily, but we have selected $L_c = 20.0$). This distribution of $L_i$ corresponds to $v = 0$ (*and is preferable for our analysis*).

Another extreme point is where $L_1 = 10.0$ and $L_i = 0, \forall i \neq 1$. This distribution of $L_i$ corresponds to $v = 1$ (*and is not preferable for our analysis*). Finally, in the case of using values $v \in (0, 1)$ the distribution of $L_i$ across clients will be linearly interpolated between these two distributions corresponding to the two cases described above.

In our experiments, we set $v = 0.1$.

**Controlling the sparsity parameter $c$.** The process of dataset generation starts with constructing a matrix $\mathbf{S}$ with $n$ rows and $d$ columns with $\mathbf{S}_{i,j} \in \{0, 1\}$. We set $\mathbf{S}_{i,j} = 1$ when client $i \in [n]$ depends on the coordinate $j \in [d]$, otherwise we $\mathbf{S}_{i,j} = 0$. The filling of $\mathbf{S}$ happens column-wise. The columns $s_j \in \mathbb{R}^n$ of the matrix $\mathbf{S}$ are filled with values 1 in positions corresponding of a random subset of $[d]$ of cardinality $c$ chosen uniformly at random. If after processing all columns there exists a client that depends on 0 coordinates, the strategy of filling is restarted. In the logic of our generation algorithm, we use 5 attempts to create a valid filling. If the dataset sparsity generation procedure fails after all attempts, we report the failure of the dataset generation process.

**Generating datasets.** Each client has a loss function $f_i(x) : \mathbb{R}^d \rightarrow \mathbb{R}$. However, due to the previous construction of the sparsity pattern, the client $i$ depends on the coordinates $\{j : \mathbf{S}_{i,j} = 1\}$. After renaming variables and ignoring variables that $f_i(x)$ does not depend on, we define $f_i(z)$ as:

$$f_i(z) \stackrel{\text{def}}{=} \frac{1}{n_i} \left\| \mathbf{A}_i \cdot x(z) - b_i \right\|^2 .$$

Next, we generate a uniform spectrum $[1.0, 20.0]$ and fill $\mathbf{A}_i$ in such a way that the spectrum $\lambda \left( \frac{2}{n_i} \mathbf{A}_i^\top \mathbf{A}_i \right)$ is represented by a linear interpolation controlled by the meta parameter $v$ from the uniform spectrum $[1.0, 20.0]$ to $[L_c, 0.0, \ldots, 0.0]$. After constructing $\mathbf{A}_i$, we set $b_i \stackrel{\text{def}}{=} \mathbf{A}_i \cdot x_{\text{solution}} + noise_i$. The $x_{\text{solution}}$ plays the role of a prior known solution, and $noise_i \sim \mathcal{U}_{[-1,1]} \cdot p$ is additive noise in the linear model, where $p \in \mathbb{R}$ is a fixed constant. In our experiments, $p = 2.0$. It plays the role of a perturbation that scales the standard deviation of the zero mean r.v. $noise_i$. It helps to escape the interpolation regime, i.e. situation in which $\nabla f_i(x^*) = 0, \forall i \in [n]$.

### C.2 LOGISTIC REGRESSION WITH ADAPTIVE STEPSIZE

In this section, we provide additional numerical experiments in which we compare EF21 under the standard analysis and our analysis. We address the problem of training a binary classifier via a logistic model on several LIBSVM datasets (Chang and Lin, 2011).

**Computing and software environment.** We used the Python software suite FL_PyTorch Burlachenko et al. (2021) to simulate the distributed environment for training. We trained logistic regression across $n = 300$ clients in the experiments below. We ran the experiments on a compute node with Ubuntu 18.04 LTS, 251 GBytes of DRAM memory, and 48 cores of Intel(R) Xeon(R) Gold 6246 CPU @ 3.30GHz. We used single precision (FP32) arithmetic.

**Experiment setup.** We conducted distributed training of a logistic regression model on A9A, MUSHROOMS, W5A, PHISHING datasets. This setting is achieved by specifying for Equation (1) the

functions $f_i(x)$ as:

$$f_i(x) \stackrel{\text{def}}{=} \frac{1}{n_i} \sum_{j=1}^{n_i} \log\left(1 + \exp(-y_{ij} \cdot a_{ij}^\top x)\right), \qquad (a_{ij}, y_{ij}) \in \mathbb{R}^d \times \{-1, 1\}.$$

The initial shifts for EF21 are $g_i^0 = 0, \forall i \in [n]$. All datasets are randomly reshuffled and spread across clients in such a way that each client stores the same amount of data points $n_i$; the residual is discarded. The initial iterate $x^0$ is initialized as $\mathcal{U}_{[-\sqrt{1/D}, \sqrt{1/D}]}$ according to the default initialization of a linear layer in PyTorch [3]. For standard EF21 we used the largest step size allowed by its theory.

**Reasons of sparse features.** In addition to what we have already mentioned in Section 4 we would like to highlight reasons of appearing rare features in the training based on our experience:

First, when the input for ML models is a categorical value from a finite set $S$, not a real number, a special conversion is needed. If $S$ has no natural order, the conversion usually maps each $s \in S$ to a one-hot vector $\tilde{s} \in \mathbb{R}^k$, where $\tilde{s}$ has only one non-zero element. This conversion has drawbacks, such as introducing an artificial partial order in $\mathbb{R}^k$. It is used for models that cannot handle categorical inputs directly without conversion. Examples of models are Neural Nets and Linear Models.

Second, some features may be inherently sparse vectors in the application. For example, if $a$ encodes a voxel grid of solid geometrical physical objects, it will be a sparse vector in most applications.

Third, during the modeling stage, there may be a specific pattern called "*feature template*", which defines how a family of close-by features is evaluated. This technique is often used in applications where ML is applied for complex tasks that require defining the input features as part of the problem, and they are not given in advance, e.g. because there is no established practice for specific tasks.

**Practical applicability of our analysis for a case when $c = n$.** The LIBVSM datasets are mostly sparse datasets. As we explained above, this is not an unrealistic assumption. However, in practice, the Definition 9 may be too strict. According to this definition, adding a bias (or intercept) term to the Machine Learning model during training leads to $c = n$. We visualize the sparsity patterns in A9A, MUSHROOMS, PHISHING, W5A in Figure 5. This representation is obtained after uniformly shuffling the original train datasets and splitting them across $n = 300$ clients.

In this experiment, we consider the setting where the Master executes Algorithm 1, but with a varying step size $\gamma^t$ in Line 4.

In our modification, we use the maximum theoretical step size from Theorem 2, but we define the parameter $c$ based on Lemma 5 because it reflects the original notion of $c$ when training Machine Learning models with intercept terms that lead to $c = n$. Moreover, we define the quantity $L_+$ based on Lemma 2. We summarize the rules for executing the adaptive version of EF21 as follows:

$$c^t \stackrel{\text{def}}{=} \frac{\|g^t - \nabla f(x^t)\|^2}{G^t/n},$$

$$L_+ \stackrel{\text{def}}{=} \min\left\{\sqrt{\frac{c^t \max_i L_i^2}{n}}, \sqrt{\frac{\sum_{i=1}^n L_i^2}{n}}\right\},$$

$$\gamma \stackrel{\text{def}}{=} \gamma^t \stackrel{\text{def}}{=} \frac{1}{L + L_+\sqrt{\frac{c^t}{n}} \frac{\sqrt{1-\alpha}+1-\alpha}{\alpha}}. \tag{37}$$

The system of rules defined in system of equations (37) raises several questions:

1. How can we estimate the quantity $c^t$? To estimate $c^t$ from this definition, we need to be able to estimate $\|g^t - \nabla f(x^t)\|^2$ in the master, but the vector $\nabla f(x^t)$ is not available in the master.

2. How can we analyze the convergence of $\Psi^t$ when $c^t$ varies? If we allow $c^t$ to change during the optimization process, then the Lyapunov function from Equation (35) also changes over time. And varying $c$ will make it very hard to analyze the behavior of $\Psi^t$.

---

[3]Information about initialization linear layer torch.nn.Linear

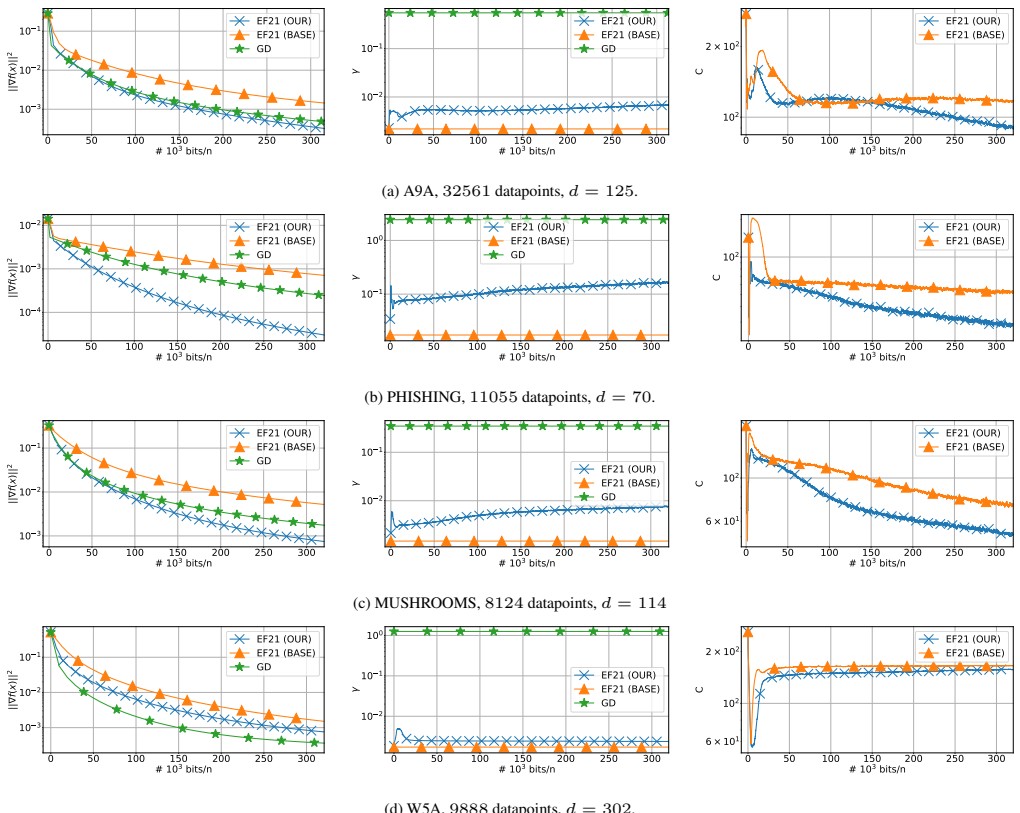

(a) A9A, 32561 datapoints, $d = 125$.

(b) PHISHING, 11055 datapoints, $d = 70$.

(c) MUSHROOMS, 8124 datapoints, $d = 114$

(d) W5A, 9888 datapoints, $d = 302$.

Figure 4: Training Logistic Regression model across $n = 300$ client. Dimension of problem $d$ is presented in the plots, $c^t$ is the behavior of modified notion of variable $c$, $\gamma$ is the value of used step size. Computation is carried out in FP32. Full client participation. The step size used for standard EF21 and GD are theoretical. Used client compressors for EF21 algorithms are Top1.

We will not address these questions in our experiment below. We believe that a deep understanding of these questions is the subject of future research. The purpose of this experiment is to demonstrate that the notion of $c^t$ can open new opportunities for research in this direction.

**Results and Conclusion.** We present the results in Figure 4. In datasets A9A (a), PHISHING (b), MUSHROOMS (c), we can increase the step size by a factor of $10x$ using our proposed scheme. In dataset W5A (d), we can only increase the step size by a small factor of $1.35x$, which suggests the need for more refined analysis. In all experiments, the EF21 with our approximate scheme performs better than the standard EF21. We do not observe any convergence or stability issues in any of the experiments. We also show the behavior of $c^t$, which is not exploited by the standard EF21. We hope this experiment will inspire future research in the direction of adaptive $c^t$.

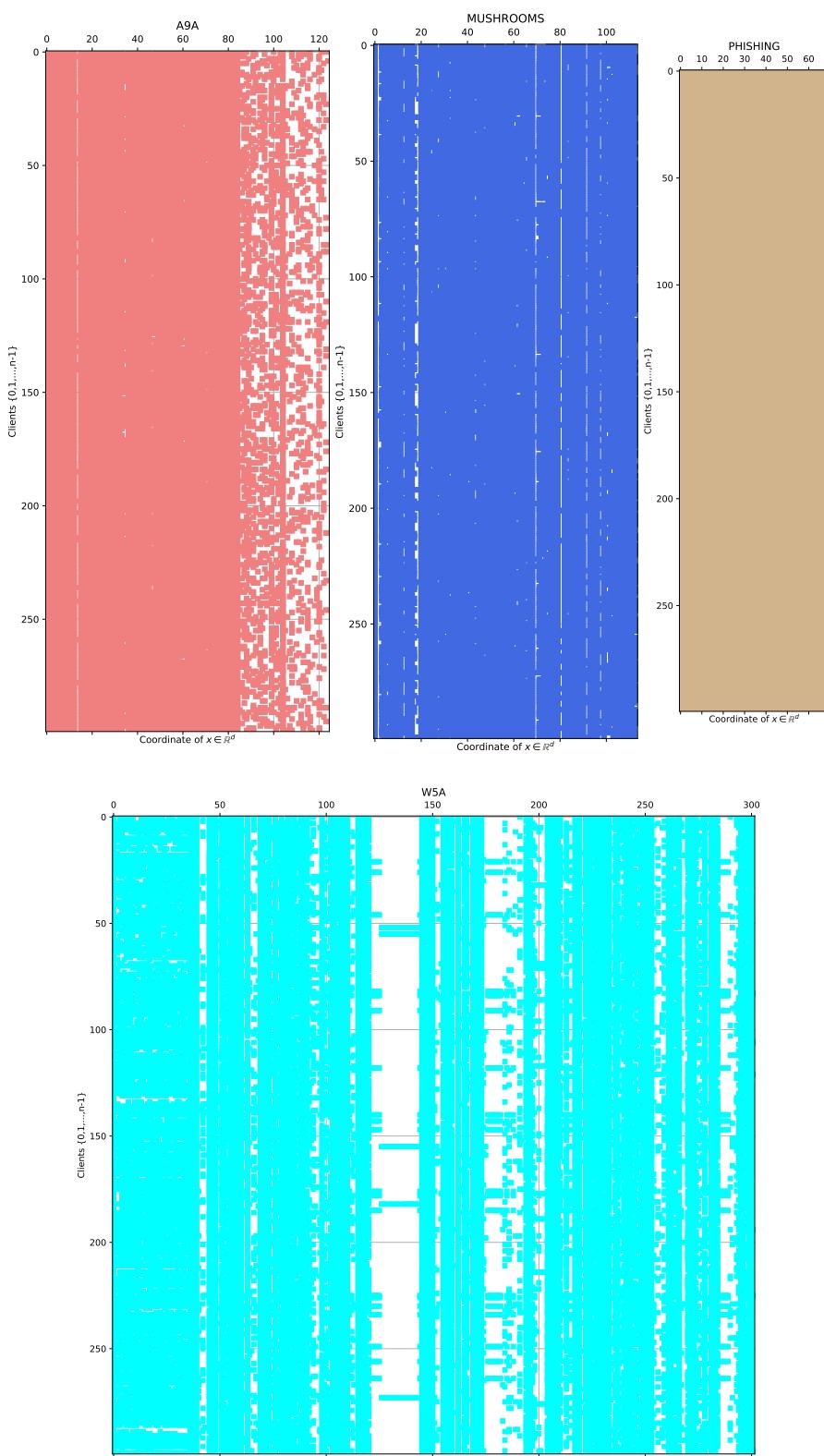

Figure 5: Sparsity patterns of datasets for training Logistic Regression model across $n = 300$ clients. An empty cell indicates that a specific client does not have any data for a specific trainable scalar variable $x_i$. The plots show the set $[n] \times [d] \backslash \mathcal{Z}'$ from Section 4.

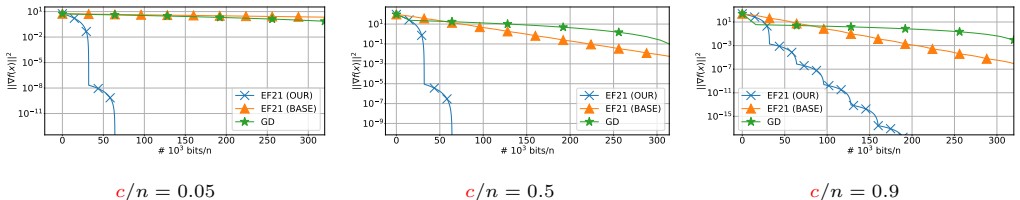

$c/n = 0.05$ $c/n = 0.5$ $c/n = 0.9$

Figure 6: Comparison of the performance of EF21 + Top1 with the standard approach proposed by Richtárik et al. (2021) and a newly proposed stepsize (see Theorem 2) on the linear regression problem and GD for non convex case. The sparsity pattern $c/n$ is changing in controlled way.

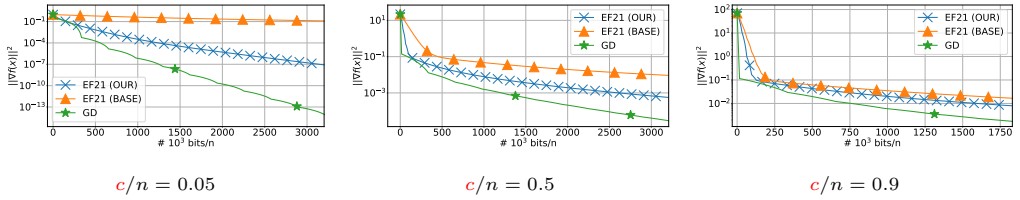

$c/n = 0.05$ $c/n = 0.5$ $c/n = 0.9$

Figure 7: Comparison of the performance of EF21 + Top1 with the standard approach proposed by Richtárik et al. (2021) and GD on the linear regression problem. The optimization objective is convex. The sparsity pattern $c$ is controlled by manipulating the sparsity of the data.

## D ADDITIONAL EXPERIMENTS

### D.1 LINEAR REGRESSION ON SPARSE DATA WITH NON-CONVEX REGULARIZATION

In this experiment, we consider the minimization of the function $f(x)$, which has the following form:

$$f(x) = \frac{1}{n} \sum_{i=1}^{n} (f_i(x) + \phi_i(x),$$

$$f_i(x) = \frac{1}{n_i} \|\mathbf{A}_i x - b_i\|^2,$$

$$\phi_i(x) \stackrel{\text{def}}{=} \lambda \cdot \sum_{j=1}^{d} \frac{x_j^2}{x_j^2 + 1} \cdot I_{ij},$$

$$I_{ij} \stackrel{\text{def}}{=} \begin{cases} 1, & \text{if } f_i(x) \text{ depends on } x_j \\ 0, & \text{otherwise} \end{cases}$$

The term $\phi(x)$ is a non-convex regularizer, with eigenvalues in the set $\left[-\frac{1}{2}, 2\right]$. We use the dataset generation algorithm described in Section C.1. It also explains the main meta-parameter that controls the generation of optimization problem instances such as $c/n$. After the generation process, the quadratic part depends only on a subset of variables $(x_1, \ldots, x_d)$. Unlike in Section C.1, we also restrict $\phi_i(x)$ to depend only on that subset of variables. This is why we include indicator variables $I_{ij}$ in the general formulation. We set the regularization coefficient $\alpha = 3 \cdot \max(L_{f_i})$ to make $f_i(x)$ non-convex smooth functions.

We present the results for varying $c/n$ in Figure 6. As we can see, $c/n$ plays a crucial role in the convergence of EF21. Specifically, as $c/n$ goes to zero, our new analysis allows us to increase the step size of EF21. As $c/n$ goes to one, the step size of EF21 becomes more similar to the standard one. We use the meta-parameter $v = 0.1$, which enforces the condition from Lemma 2 and which is favorable for our algorithm and analysis in the context of Main Theorem 2.

### D.2 EF21 VERSUS GD ON CONVEX OBJECTIVE

In Section 6.1 we have observed that we gain improvements for EF21 with our analysis compared to standard analysis. In this section, we add a comparison of these algorithms in addition to GD. For all algorithms, we have used theoretical step size. Results presented in Figure 7.

As we see GD behaves better compared to the classical analysis of EF21 and our proposed analysis if do not tune step sizes. An important caveat is that EF21 is an algorithm for non-convex optimization and this setting is convex. In conclusion, we see to apply EF21 in a convex setting with theoretical step sizes there is room for future research.

## E   LIMITATIONS

We acknowledge that the practical applicability of our results is limited because, firstly, many real-world datasets are not sparse enough to enjoy a significantly small $c/n$ ratio, and secondly, the complex architecture of deep neural networks already has non-zero weights on the second layer, independent of the initial dataset's sparsity, further increasing the value of $c$.

