# OpenReview forum: "Error Feedback Shines when Features are Rare"
_ICLR.cc/2024/Conference — Submitted to ICLR 2024_

### Official Review · Reviewer_CuB9 · 2023-10-30

**Soundness:** 4 excellent
**Presentation:** 4 excellent
**Contribution:** 3 good
**Rating:** 8
**Confidence:** 3

**Summary:**

In the paper, the authors showed that greedy sparsification (TopK compressor), together with error feedback, can beat distributed gradient descent in terms of theoretical communication complexity, by characterizing its fast convergence rate in a certain regime (that depends on the sparsity parameters c and r).
See Example 1 and Theorem 2.
Numerical experiments are provided Section 6 to validate the proposed theoretical analysis.

**Strengths:**

The paper addresses the important problem of communication complexity in distributed ML.
The
The paper is in good shape.

**Weaknesses:**

I do not see particular weakness for the paper but a few comments, see below.

**Questions:**

I do not have specific questions but the following general comments for the authors:

1. when referring to the appendix, please specify which section/part of the appendix.
2. I personally suggests the authors to further elaborate on the limitations of the analysis and future work, and move them to the main text (instead of leaving them in the appendix).

---

> ### Author Response · Authors · 2023-11-22
> **Strengths**
>
> Thank you!

---

> ### Author Response · Authors · 2023-11-22
> **Questions**
>
> We will follow your advice wrt the general comments. Thanks!

---

### Official Review · Reviewer_XpPb · 2023-10-31

**Soundness:** 2 fair
**Presentation:** 2 fair
**Contribution:** 2 fair
**Rating:** 3
**Confidence:** 4

**Summary:**

The paper studies the theoretical advantage of the error feedback mechanism for compressed distributed gradient descent (DGD). The authors first defined two quantities that measure the sparseness and rareness of the data features. Then the non-convex convergence rate shows that when the data has sparse and rare features, EF21 has better communication complexity than DGD. Some simple experiments are conducted to justify the theory.

**Strengths:**

Strength:
+ The writing is clear and easy to follow.
+ The introduction to related works and existing results is comprehensive and helpful to understand the context.
+ Introducing feature “rareness” to the analysis of optimization methods is a good attempt.

**Weaknesses:**

Weakness: There are several limitations of this work.

1.	In my opinion, the motivation for this work is not very strong.

a.	The authors claim that the goal is to explain why EF21 empirically performs much better than DGD in terms of communication complexity but theoretically does not. However, their analysis relies on some strong assumptions about the data which are uncommon in practice. Indeed, does the fact that EF21 performs well in practice on many types of data (without strong assumptions) indicate that the feature rareness and sparsity assumed in this paper are NOT the true reasons? While there are many engaging words like “breakthrough” or “milestone”, I don’t really feel surprised because a better rate shall be expected when we limit the function class and propose strict data assumptions. But I doubt whether this is the correct direction given the points above.

b.	It seems that Lemma 2 – 5 are general results not specific to EF. Can we apply the same analysis and argument to DGD? In other words, can rare and sparse features also improve the rates of DGD?

2.	It seems that the arguments are limited to simple linear models. This is because the feature sparsity would lead to model sparsity (which is a key component in the analysis, for example Lemma 5) only for linear models. For non-linear models (for example the DNNs), the arguments will not hold.

3.	Empirically, the experiments also only used convex regression models but not more complicated neural networks. There is a discrepancy between the experiments with the non-convex theoretical analysis. So, the experiments are kind of limited.

4.	As a theoretical paper, the setups and technicality are not comprehensive and strong enough. The paper only studied deterministic setting without stochastic gradients. SGD-type methods are more practical. What’s the situation in the stochastic setting? Does the same problem exist? From the technical perspective, the main modification in the proof compared with prior works is improving Young’s inequality and the smoothness constant $L$ using rareness/sparsity. This is not very challenging and novel in my evaluation.

**Questions:**

See as above

---

> ### Author Response · Authors · 2023-11-22
> **Strengths**
>
> Thanks!
>
> However, it seems you overlook one our most significant contributions. Our work is the first to introduce a theoretical framework in which Error Feedback surpasses Gradient Descent in theory (in communication complexity). No such work exists since 2014 when Error Feedback was first proposed by Seide et al.

---

> ### Author Response · Authors · 2023-11-22
> **Weakness 1a**
>
> > 1. In my opinion, the motivation for this work is not very strong.
>
> > a. The authors claim that the goal is to explain why EF21 empirically performs much better than DGD in terms of communication complexity but theoretically does not. However, their analysis relies on some strong assumptions about the data which are uncommon in practice. Indeed, does the fact that EF21 performs well in practice on many types of data (without strong assumptions) indicate that the feature rareness and sparsity assumed in this paper are NOT the true reasons? While there are many engaging words like “breakthrough” or “milestone”, I don’t really feel surprised because a better rate shall be expected when we limit the function class and propose strict data assumptions. But I doubt whether this is the correct direction given the points above.
>
> **We believe that, to the contrary to what the reviewer says, the motivation for our work is strong.** Our motivation comes from the observation that **despite nearly a decade of research into error feedback (EF14 was proposed by Seide et al in 2014), there is not a single paper which would prove theoretically that error feedback surpasses gradient descent in communication complexity in some regime.** We explain this carefully and step by step in Sec 2 and 3. We believe the motivation to explain this discrepancy is strong -- it is highly important to the future of ML to explain theoretically why and when popular ML methods & tricks work. This applies to dropout, batch norm, learning schedules, random data shuffling and many more tricks. In our case, we look at error feedback -- a technique used for almost a decade, one that is critical to the success of distributed learning.
>
> However, **we do not see any criticism in the review of our motivation.** Instead, we see a criticism of our approach / results; and we shall address them below. Please can you correct the review and mention that the motivation behind our work, as explained above and also in our paper, is very reasonable and timely? If you do not believe this to be the case, please explain why.
>
> Now we come back to our results. Indeed, you are right that error feedback works well for many datasets and regimes in practice. You are also right that not all of these are explainable by the rare features regime. **We never claim in our paper that all of the success of error feedback in practice is explained by the rare features regime. That would be a very strong and clearly unsubstantiated claim. For instance, in the abstract we say "Perhaps surprisingly, we show that EF shines in the regime when features are rare...". The logical structure of this sentence is an implication: "if the features are rare, then EF shines". We do not claim the opposite implication: "whenever EF shines, the features are rare".**
>
> Please note we argue in Sec 3 that some *homogeneity-limiting* assumptions need to be made in order for EF21 to work better than GD. We argue this in Sec 3.3, where we explain that in the homogenenous data regime, EF21 does not scale with the number of workers at all. This prevents the method to become better than GD. **So, the criticism "I don’t really feel surprised because a better rate shall be expected when we limit the function class and propose strict data assumptions" is unjustified. Indeed, it is *necessary* to limit the class of functions, steering them away from homogeneity.** Moreover, we believe this should be surprising since in virtually all of distributed training, and especially in federated learning, it is heterogeneity that causes issues, not homogeneity. So, our insight that EF does *not* work well in the homogenenous regime, which motivated our search for a suitable heterogeneity regime friendly to error feedback, can be seen as counterintuitive and surprising. We managed to identify the rare features regime as a *sufficient* heterogeneity assumption under which we can formally show that EF21 outperforms GD in communication complexity. We stress the word "sufficient" since we do not claim it is necessary. Perhaps, motivated by our work, other "friendly-to-EF" heterogeneity regimes will be identified by others in the future. In fact, we strongly believe this is precisely what will happen.
>
> **When used in our paper, the words "breakthrough" or "milestone" have a very specific meaning and context. Since our work is indeed the first to identify a regime in which error feedback outperforms GD in theory after nearly a decade of research into this topic, we believe these words are justified.** Having said that, our work is not the end of the story. Feature rareness is merely sufficient and not a necessary assumption for error feedback to work well. In this sense you are of course right when you say "feature rareness and sparsity assumed in this paper are NOT the true reasons" (it would be more appropriate though to replace the word "true" by "only"). But we never claimed otherwise.
>
> We hope this settles your concerns!

---

> ### Author Response · Authors · 2023-11-22
> **Weakness 1b**
>
> > b. It seems that Lemma 2 – 5 are general results not specific to EF. Can we apply the same analysis and argument to DGD? In other words, can rare and sparse features also improve the rates of DGD?
>
> The claim that Lemmas 2-5 are general results not specific to EF is not accurate. We now comment on these results:
> - Lemma 2 is new and potentially useful in other contexts (we do not know where though), but is not used in the analysis of DGD since DGD does not seem to benefit from Assumption 4. Indeed, the quantity $L_+$ appearing there is always larger than $L$ (the smoothness constant of $f$), and DGD depends on the better constant $L$ already (see the first line of Table 1, for example). We use Lemma 2 to analyze EF21 in the rare features regime since the quantity $L_+$ does appear in the analysis of EF21.
> - Lemma 3 is clearly related to the contraction properties of the TopK compressor (used in EF21, for example), and hence has nothing to do with DGD, which does not compress gradients at all.
> - Lemmas 4 and 5 are also *not* used in the analysis of DGD, and we do not see how these lemmas would be relevant. For example, in DGD, $g^t = \nabla f(x^t)$, and hence the left-hand  and right-hand sides of (15) would be always zero, independently of the quantity $c$ which is central to our analysis of EF21. Lemma 4 supports Lemma 5.
>
> However, it's important to note (and we do so in the paper) that DGD's rate is influenced by the quantity $r$, as detailed in Table 1 of our paper (see also the abstract).

---

> ### Author Response · Authors · 2023-11-22
> **Weakness 2**
>
> > 2. It seems that the arguments are limited to simple linear models. This is because the feature sparsity would lead to model sparsity (which is a key component in the analysis, for example Lemma 5) only for linear models. For non-linear models (for example the DNNs), the arguments will not hold.
>
> Our results apply to any setup when the assumptions of our theory apply. Whenever the "feature sparsity" coefficient $c$ defined in (9) is small, we get benefits. The smaller it is, the more theoretical benefits we get (but the theory works also if it is large). This assumption is in no way limited to linear models (but it is true it holds for generalized linear models with sparse data).
>
> What we are going to mention next we did not explicitly comment on in the paper, but we will add this as a highly important example of a situation in which our results are highly relevant.
>
> Training under **model heterogeneity:**
>
> **Note that $c$ is small in situations when each client is merely trying to learn a small submodel of a very large global model $x$ (consisting of $d$ real parameters) due to local memory/capacity issues. This makes sense when the global model is so large that it can't be stored on any of the machines, each of which has a limited memory, and if each global parameter $x_i$ is being learned by at most $c$ clients, where $c$ is small.**  This model-heterogeneous approach to distributed training is increasingly popular and important in the training of very large models, i.e., in "model parallel/heterogeneous" training. See
> - Dashan Gao, Xin Yao, and Qiang Yang. A survey on heterogeneous federated learning. arXiv preprint arXiv:2210.04505, 2022,
> - Yuang Jiang, Shiqiang Wang, Victor Valls, Bong Jun Ko, Wei-Han Lee, Kin K Leung, and Leandros Tassiulas. Model pruning enables efficient federated learning on edge devices. IEEE Transactions on Neural Networks and Learning Systems, 2022
> for works on model-heterogenenous distributed traning.
>
> **In summary, we wish to stress that whether or not $c$ is small is not necessarily related to whether the model we are trying to learn is linear or not.**

---

> ### Author Response · Authors · 2023-11-22
> **Weakness 3**
>
> > 3. Empirically, the experiments also only used convex regression models but not more complicated neural networks. There is a discrepancy between the experiments with the non-convex theoretical analysis. So, the experiments are kind of limited.
>
> Please note that in our work we do *not* propose a new method. Error feedback has been known and successfully used for nearly a decade. For example, it was used in "CocktailSGD: Fine-tuning Foundation Models over 500Mbps Networks, ICML 2023" to obtain some impressive empirical results wrt distributed fine-tuning of LLMs. So, our aim is not to show that EF (whether in EF14 or EF21 forms) works well - this is known.
>
> Our goal is to prove for the first time that error feedback can beat GD in theory as well. We prove that this happens in the sparse features regime (which, as explained in a separate post, also holds for model-heterogenous training scenarios!). So, our contribution is theoretical, and large scale experiments are not really needed. Our experiments, while not very large, are designed to show that our theoretical predictions indeed translate into gains when the method is coded up and executed.
>
> **Having said that, we will add an experiment with larger NNs in the context of model-heterogeneous training. This will take us some time to prepare - we will certainly have this ready by the camera-ready deadline.**

---

> ### Author Response · Authors · 2023-11-22
> **Weakness 4**
>
> > 4. As a theoretical paper, the setups and technicality are not comprehensive and strong enough. The paper only studied deterministic setting without stochastic gradients. SGD-type methods are more practical. What’s the situation in the stochastic setting? Does the same problem exist? From the technical perspective, the main modification in the proof compared with prior works is improving Young’s inequality and the smoothness constant $L$ using rareness/sparsity. This is not very challenging and novel in my evaluation.
>
> We studied the deterministic setting since the problem we study is already open in the canonical deterministic setting. One needs to resolve it here before moving on to the stochastic setting. However, we see no fundamental difficulty in extending our results to the stochastic setting. We can include such results in the appendix of the camera ready version of the paper.
>
> In retrospect, our analysis may not *look* very challenging to some people. We can't influence this. But this is because we eventually found what we believe is a beautiful and simple solution to the problem we set out to solve. It was not clear at all what to do at the start of our research journey, which took us many months to complete. We had various much more complicating-looking but much less theoretically striking results before we managed to find a simple solution.
>
> **We would thus very much welcome if the reviewer could decide to appreciate the beauty that can be found in simplicity.  Also, we would appreciate if the reviewer could judge our works by the results we obtain. We believe the result is strong. Sometimes strong results are obtainable using relatively simpler insights, sometimes heavy machinery needs to be used. This depends on the nature of the problem. It turns out that the nature of the problem we set out to solve was amenable to a cute and simple solution. Please note that often it is much more difficult to find a simpler solution to a problem than a more difficult-looking (but also a more convoluted) one.**
>
> Thank you!

---

### Official Review · Reviewer_xzAm · 2023-10-31

**Soundness:** 4 excellent
**Presentation:** 4 excellent
**Contribution:** 3 good
**Rating:** 8
**Confidence:** 4

**Summary:**

This paper studies error feedback in distributed optimization. It provides a first theoretical analysis of how greedy sparsification and error feedback can improve the communication complexity of distributed gradient descent. Specifically, when $\sqrt{\frac{c}{n}}L_+ \leq L$, the communication complexity improves. Numerical experiments are conducted to validate the theoretical results.

**Strengths:**

1. This paper is well written and most ideas are presented in a straightforward and easy-to-read manner.
2. It provides meaningful observations to motivate the research.
3. Theoretical results are solid and only rely on simple and standard assumptions.

**Weaknesses:**

1. I have some concerns about the novelty and contribution of the paper since it mostly builds on the previous work EF21. I hope the authors can clarify this and explain how this work advances error feedback algorithms or distributed optimization in the future.
2. As the authors said in the paper, the experiments are rather toy and the practical applicability is limited because most real-world datasets are not sparse enough. It would be helpful and convincing to conduct some experiments on real-world datasets.

**Questions:**

in the above

---

> ### Author Response · Authors · 2023-11-22
> **Strengths**
>
> Thanks for the nice comments!

---

> ### Author Response · Authors · 2023-11-22
> **Weaknesses**
>
> > 1. I have some concerns about the novelty and contribution of the paper since it mostly builds on the previous work EF21. I hope the authors can clarify this and explain how this work advances error feedback algorithms or distributed optimization in the future.
>
> We believe our paper already gives an answer to this question at length in Sections 2 and 3. Here we give a very brief summary: Error feedback was first proposed in 2014 by Seide et al (we call it EF14). It is heavily used in distributed training since it is empirically very well performing. However, EF14 had some serious theoretical issues - its theoretical communication complexity is worse than that of vanilla gradient descent, moreover, this worse complexity is obtained using stronger assumptions than those needed to analyze GD. This is where EF21 steps in: it is a new variant of error feedback which at the same time improves upon empirical performance of EF14 *and* fixes these theoretical issues. While, EF21 offers the current (prior to our work) SOTA empirical performance and theoretical guarantees (for error feedback methods), the theoretical guarantees are still at best the same as those of GD. Our work builds on EF21 because this is the current SOTA in practice and theory. Our work is the first to identify a regime, which we call the sparse features regime, in which any form of error feedback is provably shown to outperform GD. In particular, we did this for EF21. We do not know how to do this for EF14 since the best rates for EF14 are still worse than those of GD.  We believe that our work will inspire others to identify new regimes in which this is the case.
>
> > 2. As the authors said in the paper, the experiments are rather toy and the practical applicability is limited because most real-world datasets are not sparse enough. It would be helpful and convincing to conduct some experiments on real-world datasets.
>
> We used synthetic datasets since in these we can precisely control the sparsity parameters $c$ and $r$ which show up in our theory. This way, we can check whether our theoretical predictions translate to appropriate gains in an actual run of the algorithm. Our experiments conclusively show that whenever our theoretical assumptions apply, our theoretical predictions indeed translate into empirical gains. Therefore, we believe there is no reason to doubt that whenever the parameter $c$ is small for any dataset practitioners care about which we currently do not have access to, EF21 would indeed benefit, as our theoretical and empirical evidence on synthetic datasets clearly show. We have experiments with more real-world datasets in Section C.1. However, we can add experiments on other datasets as well.

---

> > ### Comment · Reviewer_xzAm · 2023-12-05
> >
> > Thank you for addressing my concerns!

---

### Official Review · Reviewer_h2wk · 2023-11-01

**Soundness:** 3 good
**Presentation:** 3 good
**Contribution:** 3 good
**Rating:** 6
**Confidence:** 3

**Summary:**

In this paper the authors try to make sense on the gap in our understanding of the theoretical and practical aspects of gradient descent algorithms in the distributed setting. The try to reason why the algorithm based on heuristics like the greedy sparsification and error feedback performs better in practice than the distributed gradient descent, but theoretically the opposite is observed. They identify scenarios when "features are rare" and prove that in these scenarios one can prove that the performance of the heuristic algorithms are better than the distributed gradient descent.

**Strengths:**

This is one of the few papers trying to understand, or rather prove theoretically, why is a heuristic algorithm performing better than another algorithm in practice though theory suggests otherwise.

**Weaknesses:**

The paper proves that under certain assumptions the EP21 algorithm performs better than the DGD algorithm. The assumptions are quite strong and hence it is not clear if this is best scenario to explain the performance of EP21 algorithm.

**Questions:**

Can you say how often one expects to find real life data satisfying the assumptions under which the improved theoretical study is done?

---

> ### Author Response · Authors · 2023-11-22
> **"it is not clear if this is best scenario to explain..."**
>
> If there were other scenarios explaining why error feedback works better than GD in practice, the question of *which* scenario is the best would make sense. However, no such scenario was identified in any work we know of, and our "sparse features" regime is the first and only known regime in which this happens. We believe that in this sense, our work is a breakthrough. We hope that following our work, more such scenarios will be discovered by others in the future, and that the community will in this way progressively learn more about why error feedback works much better than GD.
>
> We do not know how to answer this question *quantitatively* since we are not in possession of a lot of real world data, which is often proprietary and not available to researchers.  We certainly do not claim that our work applies to all datasets and situations. It does not. However, there are real-world scenarios where the sparse regime indeed is present. A short discussion of why sparse feature patterns may arise in practice can already be found in Appendix C.2 in the paragraph “Reasons of the sparse feature”.

---

### Meta-Review · Area_Chair_bEDR · 2023-12-05

**Metareview:**

This paper studies distributed optimization of a finite sum of smooth nonconvex functions. The starting point of the paper is an observation that the error feedback algorithm EF14/EF21 often outperforms distributed gradient descent in practice, but their theoretical bounds are comparable. The goal of this paper is to provide a refined theoretical justification on the performance of EF14/21. To this end, the paper studies a regime where features are sparsely distributed, i.e. each client only operates on a portion of the features, making the gradient matrix sparse. Such sparsity pattern leads to a improved convergence rate where sparsity as a parameter enters.

**Strength**
- The paper is clearly written and well-motivated. The literature review is thorough.
- The assumptions and main results are clearly explained.
- The convergence rate is better than distributed gradient descent when features are sparse.
- Some numerical results are presented to test the theory.

**Weaknesses**
- The main concern on the paper is about the technical novelty. It turns out that the improvement in the convergence rate comes from the newly assumed sparsity pattern, but at a technical level, it is not hard to leverage such additional assumption into the state-of-the-art work from EF21 (Richtarik et al 2021) to obtain the claimed bound: one only needs to be careful in algebraic calculations and replaces steps involving dense features in EF21 with inequalities about sparse features - and such calculation, while requiring efforts (see Lemmas 2-5), cannot justify the novelty and technical contributions.

- Another concern is about the disconnection between the theory and the experiments. While the theory is established for smooth nonconvex functions, models tested in the experiments are either linear regression or logistic regression.

**Suggestions to the authors**
- Authors are suggested to add new experiments that can validate the theory on nonconvex models.
- Authors are encouraged to develop a deeper theory to better explain EF14/21, or develop new algorithms by leveraging new optimization techniques.
- It is still interesting to consider sparse gradient matrix, but authors are encouraged to think at a more insightful perspective.

**Justification For Why Not Higher Score:**

The weaknesses cannot be addressed without a major revision of the paper. For example, to address technical novelty, authors may have to develop new algorithms/theory.

**Justification For Why Not Lower Score:**

N/A

---

### Decision · Program_Chairs · 2024-01-16

Reject